# Radiant Triangle Soup with Soft Connectivity Forces for 3D Reconstruction and Novel View Synthesis

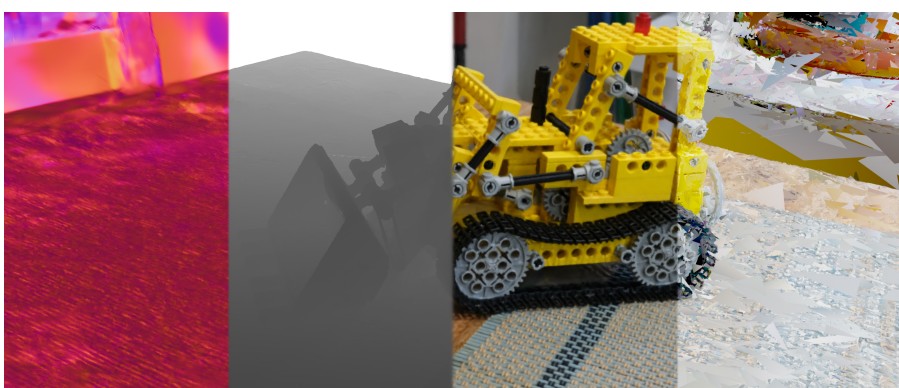

Figure 1: Optimizing Radiant Triangle Soup (RTS) produces high quality 3D models from captured images. Above, from left to right, are the rendered normal map, rendered depth map, rendered image, and the direct rasterization of the triangle primitives, as if they were opaque.

## ABSTRACT

We introduce an inference-time scene optimization algorithm utilizing triangle soup, a collection of disconnected translucent triangle primitives, as the representation for the geometry and appearance of a scene. Unlike full-rank Gaussian kernels, triangles are a natural, locally-flat proxy for surfaces that can be connected to achieve highly complex geometry. When coupled with per-vertex Spherical Harmonics (SH), triangles provide a rich visual representation without incurring an expensive increase in primitives. We leverage our new representation to incorporate optimization objectives and enforce spatial regularization directly on the underlying primitives. The main differentiator of our approach is the definition and enforcement of soft connectivity forces between triangles during optimization, encouraging explicit, but soft, surface continuity in 3D. Experiments on representative 3D reconstruction and novel view synthesis datasets show improvements in geometric accuracy compared to current state-of-the-art algorithms without sacrificing visual fidelity.

## 1 INTRODUCTION

Gaussian Splatting (GS) methods are effective at Novel View Synthesis (NVS). However 3D Gaussian Splatting (3DGS) (Kerbl et al., 2023) fails to accurately model the geometry of scene surfaces. 3D Gaussians are by design smooth, unbounded volumetric primitives, which are inherently ill-suited for representing flat surfaces and sharp boundaries. Optimization for novel view synthesis involves alpha-blending several overlapping Gaussians, none of which have to be located on the underlying surfaces. As a result, optimizing a representation of 3D Gaussian kernels for image synthesis commonly causes floaters or blurry artifacts in the scene.

Several authors (Huang et al., 2024; Guédon & Lepetit, 2024; Chen et al., 2024a; Dai et al., 2024) have proposed flattening the kernels, with either strict 2D Gaussians (Huang et al., 2024) or flattened 3D Gaussians via loss regularization (Guédon & Lepetit, 2024; Chen et al., 2024a), to better model thin surfaces. These methods have shown that modifying the underlying primitives and enforcing

optimization objectives on rendered geometry leads to improvements in geometric accuracy. While flat primitive help with surface alignment, diffuse ellipsoids still fail to accurately model depth discontinuities (see Fig. 8).

Triangles are the fundamental primitives in computer graphics because they are necessarily planar and convex. They can approximate any surface as piecewise planar at any tolerance level, can directly model sharp discontinuities, and can be rendered rapidly without approximations.

We present a new methodology for 3D reconstruction, which we named Radiant Triangle Soup (RTS) that enables gradient-based optimization of a Radiance Field (RF) using translucent triangle primitives as the scene representation (see Fig. 1). We provide a complete framework, including differentiable mechanisms for rasterization, as well as non-differentiable mechanisms for initialization, pruning, and densification. Our experiments show RTS achieves competitive results in terms of both appearance and geometry (see Fig. 7).

Conceptually, RTS provides a feature that is not supported by any other GS scene representation: an avenue for explicit information sharing among 3D among primitives. Conventional Gaussian kernels interact via alpha-blending when rendered onto common pixels. Back-propagating from the loss at each pixel to the primitives is the only means of coordination across Gaussians. Using multiple images from various viewpoints gives rise to several indirect constraints on the primitives, but direct constraints are currently absent.

Conversely, RTS is the first framework to enable direct information exchange among neighboring primitives. We formulate connectivity losses between neighboring triangle edges, allowing for a more direct and effective coordination of, and constraint on, primitive behavior throughout the optimization process. Even though using triangles as primitives alone achieves better geometric accuracy compared to Gaussian surfels, encouraging connectivity during optimization leads to more accurate surfaces with less floating artifacts (see Table 3).

Color expressivity on a per-primitive basis is another dimension where the RTS representation excels. It is common practice to use a single color parametrization for each primitive. However, this may require large numbers of overlapping primitives in order to reconstruct details in the surfaces of the scene. Using triangles as the representation allows for each vertex to encode a separate color (in the form of Spherical Harmonics), leading to more expressive primitives through bilinear interpolation.

To summarize, the work presented here:

- Develops a new scene representation through alpha-blending of triangle primitives.
- Introduces explicit 3D forces between primitives to encourage soft connectivity.
- Increases primitive expressivity via multi-color encoding.

## 2 RELATED WORK

We begin this section with Gaussian splatting formulations that favor surfaces and continue with methods that rely on non-Gaussian primitives. Surveys on other aspects of Gaussian splatting, omitted due to space limitations, include (Bao et al., 2025; Dalal et al., 2024; Luo et al., 2024; Wu et al., 2024). The seminal work of Kerbl et al. (2023) on 3D Gaussian Splatting introduced an explicit alternative to NeRF (Mildenhall et al., 2020) that is able to achieve high-quality rendering at much higher speed. 3DGS relies on interleaved differentiable optimization and non-differentiable density control of the explicit representation. The optimization process decreases view synthesis errors for one of the training images at each iteration and density control guides the placement of primitives.

2D Gaussian Splatting (2DGS) (Huang et al., 2024) modified 3DGS to prioritize the reconstruction of surfaces, rather than volumetric material. This was accomplished by collapsing the 3D GS to 2D discs by setting the minimum eigenvalue of the Gaussian to 0. The authors of SuGaR (Guédon & Lepetit, 2024) and PGSR (Chen et al., 2024a) devised similar techniques for aligning 2D Gaussian with the surfaces via regularization and a multi-view loss, respectively. Gaussian Surfels (Dai et al., 2024) rely on monocular surface normal estimates and a normal-depth consistency loss. Gaussian Opacity Fields (GOF) (Yu et al., 2024) extract surfaces as the zero-level set of 3D Gaussians. 3D-Half-Gaussian Splatting (3D-HGS) (Li et al., 2024) enables the representation of perfectly planar surfaces and hard

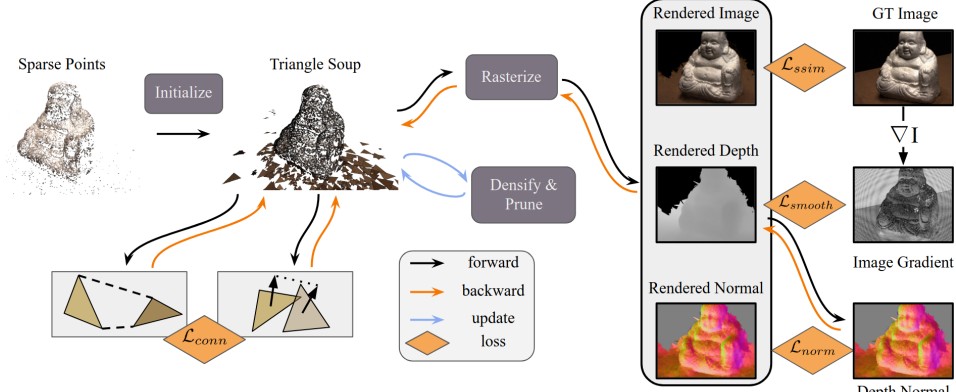

Figure 2: We begin with a set of sparse points. From these points we initialize a set of triangles and compute nearest neighbor connections for each triangle edge. During optimization, we render an image, depth map, and normal map for each view. We compute both 2D loss (over all output renderings), as well as 3D loss (directly over the primitive connections). Adaptive densification is non-differentiable and is performed at set intervals throughout the optimization. Triangle edge neighbors are recomputed following adaptive densification.

edges by attaching a splitting plane to each Gaussian, aligning it to the local surface, and setting the density of one of the half-spaces to 0.

Representations based on NeRF and 3DGS have found great success, but have also inspired researchers to seek alternatives. NeuRBF (Chen et al., 2023) utilizes Radial Basis Functions (RBFs) to overcome limitations of NeRF due to the global nature of its MLP and features. GES (Hamdi et al., 2024) is based on an explicit representation which replaces the Gaussian kernel with a Generalized Exponential Function, overcoming the low-pass effect of the Gaussian and thus requiring fewer primitives to represent the scene. Similar approaches based on smooth, non-Gaussian kernels include DARB-Splatting (Arunan et al., 2025), SolidGS (Shen et al., 2024), Beyond Gaussians (Chen et al., 2024b) and Deformable Beta Splatting (Liu et al., 2025).

Besides the methods that force their Gaussians to be planar (Huang et al., 2024; Guédon & Lepetit, 2024; Chen et al., 2024a), but still diffuse, there are others that represent the scene with collections of planar primitives. Zanjani et al. (2025) initially use Gaussian splats to model the scene and then merge them into 3D planes, which are abundant in indoor scenes. PlanarSplatting (Tan et al., 2025) is also designed for indoor scenes using planes, initialized via monocular depth estimation, as the only primitives. TRIPS (Franke et al., 2024) is based on the principle that point primitives can be rasterized into an image pyramid from which the appropriate layer can be selected according to the size of the projected point. Holes can be filled by a small network yielding accurate, crisp renderings. Triangle Splatting (Held et al., 2025a) uses triangles with diffuse edges as primitives, but does not support any mechanism for them to interact directly with each other.

Non-planar primitives were introduced by BG-Triangle (Wu et al., 2025) which uses Bézier Gaussian triangles that are effective near boundaries, but comes at the cost of operating on complex, non-planar primitives that are hard to render. Quadratic Gaussian Splatting (Zhang et al., 2025) uses deformable quadratic surfaces as primitives and geodesic, instead of Euclidean, distance-based density distributions that adapt to the curvature of the primitives.

Another class of methods rely on volumetric primitives. LinPrim (von Lützow & Nießner, 2025) is based on linear solid primitives with triangular faces and performs volumetric rendering by computing the entrance and exit points of each ray through the primitives. 3D Convex Splatting (3DCS) (Held et al., 2025b) was inspired by the limitation of GS that requires very large numbers of primitives to model hard edges and flat surfaces due to the diffuseness of the Gaussians. As the name suggests, the scene is represented by a set of polyhedral convexes which undergo volumetric rendering, pruning and splitting operations. The concepts of smoothness and sharpness introduced by 3DCS have inspired our diffuseness (see Section 3). Radiant Foam (Govindarajan et al., 2025) enables modeling light transport phenomena, like reflection and refraction, by tessellating the space into Voronoi cells and iteratively optimizing the positions of the Voronoi vertices. All methods in this paragraph

rely on different forms of volumetric elements with planar faces. **Computing the intersections of each primitive with a ray requires multiple ray-triangle intersections, compared to the single ray-triangle intersection required by RTS.**

A recent trend in the literature has been joint optimization of two representations: one for synthesizing novel views and one that is more faithful to the surfaces (Choi et al., 2024; Jiang et al., 2025). MILo (Guédon et al., 2025) maintains Gaussians that are alpha-blended for view synthesis and a watertight mesh without texture that captures the geometry of the scene.

A few common themes emerge by analyzing the above methods. Most advocate the use of bounded primitives to enhance their ability to represent sharp edges and flat surfaces with small numbers of primitives. Our primitives are bounded but have diffuse boundaries to facilitate optimization. No other method, however, has a mechanism for direct inter-primitive communication like RTS.

## 3  METHOD

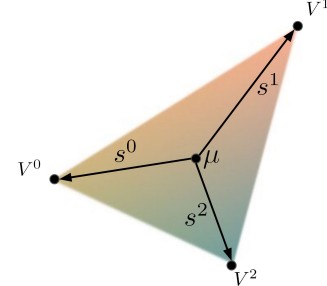

Given a set of images together with camera poses and a set of sparse points $\mathcal{S}$ computed via Structure-From-Motion (Schönberger & Frahm, 2016), we construct an explicit scene representation using triangle primitives, the parameterization and initiliazation of which we discuss in Sections 3.1 and 3.2, respectively. The triangles are endowed with diffuse boundaries, similar to 3DCS (Held et al., 2025b) (Section 3.3). To model the surfaces in the scene, we render into each camera the image, depth, and normal map of the triangles through alpha-blending (see Section 3.4). The triangle parameters are directly updated via back-propagation after computing losses between the rendering of the scene and the ground truth image from each view. We also include 2D losses on the rendered depth and normal maps (see Section 4). In our representation, primitives maintain *soft* connectivity with their neighbors, discussed in Section 3.5, for which we com-

Figure 3: Triangle parameterization. Each triangle is parameterized by the incenter $\mu$, and three scales $\begin{bmatrix} s^0 & s^1 & s^2 \end{bmatrix}$. These parameters, along with the rotation matrix $R$, define the coordinates of each vertex $V^j$.

pute additional loss directly over the connection orientations (see Section 4). Similar to previous works (Kerbl et al., 2023; Huang et al., 2024; Guédon & Lepetit, 2024; Chen et al., 2024a; Yu et al., 2024), we develop a strategy for adaptive density control in order to facilitate the addition and removal of primitives in the scene (see Section 3.6). We show an overview of our algorithm in Fig. 2.

### 3.1  PARAMETERIZATION

From the set $\mathcal{S}$ of sparse points, we first create an initial set of triangles $\mathcal{T}$. The triangle primitives $t_n \in \mathcal{T}$ in our scene representation are parametrized with,

$$t_n = \{\mu, \Delta, s, R, \alpha, \sigma\} \tag{1}$$

where $\mu$ is the incenter of the triangle, $\Delta = \begin{bmatrix} \delta^0 & \delta^1 & \delta^2 \end{bmatrix}$ are the per-vertex Spherical Harmonics, $s = \begin{bmatrix} s^0 & s^1 & s^2 \end{bmatrix}$ are the scales, $R$ is the $3 \times 3$ rotation matrix, $\alpha$ is the opacity, and $\sigma$ is the diffuse scalar, discussed in Section 3.3. **This parameterization uniquely defines each triangle primitive.**

### 3.2  INITIALIZATION

The coordinates of each sparse SfM point are used as the initialization for the incenter, and the point colors as the initialization for the zero-component of the Spherical Harmonics for all three vertices, which are then optimized separately. Similar to previous works (Huang et al., 2024; Kerbl et al., 2023), the scales for our primitives are computed based on the average distance to the three nearest neighboring points. Each triangle starts as equilateral, using the scales to parameterize the distance of each vertex from the incenter of the triangle $\mu$ along the bisectors of the angles (see Fig. 3). The diffuse scalar $\sigma$ is also set as a function of the average distance to the three nearest neighboring points **(Please see Eq. S.2). The rotation of each triangle is initialized with a random rotation matrix.**

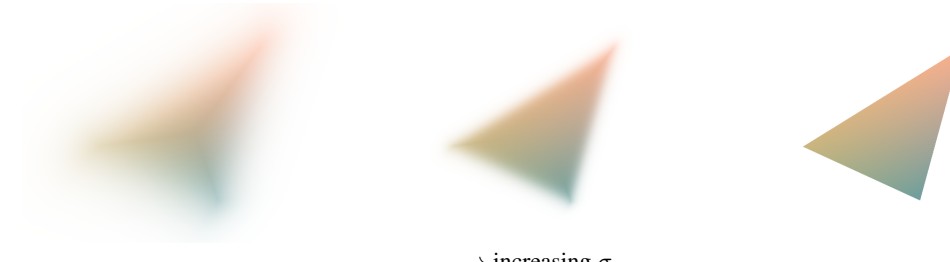

$\longrightarrow$ increasing $\sigma$

Figure 4: Controlling triangle diffuseness with $\sigma$. Small values of $\sigma$ **(left)** result in diffuse triangles with extended range of influence. Large values of $\sigma$ **(right)** result in sharp triangles with no blurring across edges.

### 3.3 DIFFUSE PRIMITIVE BOUNDARIES

Alpha-blending is the primary driving force of optimization for all splatting frameworks. Blending across primitives facilitates optimization via gradient descent and encourages, among other behaviors, movement in primitive position and orientation. It is challenging in practice to optimize a scene with fully opaque primitives that do not smoothly blend with one another. It is therefore desirable to make the triangle primitives diffuse near the edges. Taking inspiration from 3DCS (Held et al., 2025b), we parameterize the response from each triangle as a function of the signed distance from the ray-triangle intersection to the nearest edge,

$$w_\sigma = \frac{1}{1 + e^{(\sigma l)}} \qquad (2)$$

where $w_\sigma$ is the diffuse weight, $l$ is the signed distance between the intersection and the nearest triangle edge in the plane of the triangle, and the scalar $\sigma$ is an optimizable parameter of each triangle that controls the level of diffuseness. As $\sigma$ increases, the boundary of the triangle becomes less diffuse, creating sharper primitive renderings (see Fig. 4). For the signed distance, $l < 0$ occurs when the intersection point lies outside of the triangle boundary. This formulation slightly deviates from that of 3DCS, as the signed distance is computed directly, compared to their smooth approximation.

### 3.4 RASTERIZATION

Throughout this paper, we use barycentric coordinates, $\lambda = [\lambda^0, \lambda^1, \lambda^2]^T$, to rasterize triangles. Please see Section S.1 for more details. With the barycentric coordinates of the ray-triangle intersections and the diffuse weight computed, we interpolate the color for the current pixel w.r.t. the colors of each vertex weighted by the barycentric coordinates, $c_n = \begin{bmatrix} c^0 & c^1 & c^2 \end{bmatrix} \lambda$, where $c^0$, $c^1$, and $c^2 \in \mathbb{R}^3$ are computed from the SH components of each vertex, and $\lambda$ are the barycentric coordinates of the intersection point.

We aggregate the contribution to the current output pixel, $i$, from each intersected primitive, $c_i = \sum w_n c_n$, where $w_n = \alpha w_\sigma T$, and $T = \Pi_{j=1}^{i-1}(1 - \alpha_j)$ is the transmittance for the current primitive (Kerbl et al., 2023).

Previous works (Huang et al., 2024; Chen et al., 2024a; Yu et al., 2024) provide two methods for rendering per-pixel depth; (1) computing the average weighted intersection depth of all traversed primitives (mean depth), and (2) using only the depth of the primitive that causes the transmittance $T$ to exceed $0.5$ (median depth). In our work, we use median depth. Using the mean depth encourages the formation of many translucent layers of primitives. Rendering median depth removes this blending and helps guide the formation of surfaces. We directly use $d$ from Eq. S.2 for the depth of each intersection.

Additionally, we compute the surface normal, $\widehat{n}_n$, for each triangle as the cross-product between two edges. Unlike the rendered depth, to render per-pixel surface normals, we follow previous work (Huang et al., 2024; Chen et al., 2024a) and alpha-blend all the surface normals of all intersected primitives. The intuition is that through a normal consistency loss (see Section 4), the blended normals must align with the normals computed from the median depth map. This encourages all

the intersected triangles to align (and eventually collapse onto) the median surface. Thus, we render per-pixel surface normals through alpha-blending, $\overrightarrow{n_i} = \sum w_n \widehat{n}_n$.

## 3.5 CONNECTIVITY

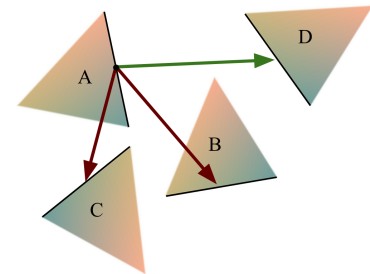

To enable soft connectivity among primitives and encourage flat, connected surfaces, we add a connectivity term to the optimization objective. For each triangle edge, we assign, and periodically update, a connection to the nearest neighboring triangle edge. Taking inspiration from the energy functions introduced in 3D scene flow estimation (Vogel et al., 2015), the connectivity term of the loss increases according to the distance between their vertices and the inner product between their normals, discussed in Section 4.

Figure 5: Examples of edge association for connecting nearby triangles. For simplicity, it is assumed the triangles are *co-planar* in this figure.

Naively connecting with the nearest edge without considering the relative orientations may cause connections that would require large changes in rotation to either triangle, leading to undesirable behaviors during optimization. To prevent this behavior, connections are only established if the triangle edges are "facing" each other. In practice, we use the inner product between the unit vectors orthogonal to the triangle edges (in the plane of each triangle) as the criterion for establishing connections. We provide an example in Fig. 5 where a connection to the highlighted edge of triangle D is valid, while the other two connections would cause large rotations.

## 3.6 ADAPTIVE DENSITY CONTROL

We perform adaptive density control through the process of cloning, splitting and pruning triangles. Previous works (Kerbl et al., 2023; Huang et al., 2024; Guédon & Lepetit, 2024; Chen et al., 2024a) perform the cloning and splitting procedures by duplicating primitives conditioned on scale and position gradients. In order to properly split large triangles, we must split them into four sub-triangles. During early iterations in the optimization, when triangles are split, the sub-triangles move independently to better align with surfaces in the scene. During the later stages when most of the surfaces have formed, the triangles split to enable more detailed surface color representation, remaining relatively attached via the connectivity forces. See Fig. 6 for a visualization of the splitting procedure.

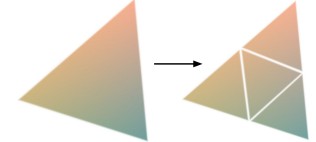

Similar to previous work, we directly clone small primitives selected for densification. Much like 2DGS (Huang et al., 2024), our method does not directly rely on the gradient of the projected 2D primitive center. Instead of computing an approximation via projecting 3D gradients into the camera plane (Huang et al., 2024), we directly condition densification on the magnitude of the incenter gradient $\nabla \mu$ in 3D.

Figure 6: Triangles are *split* with interpolated colors per-vertex.

To remove uninformative triangles, we prune primitives that meet the following criteria: (i) triangles that are transparent ($\alpha < 0.05$), (ii) triangles with one or fewer edge connections, (iii) triangles that do not intersect the camera frustum of at least three views after each epoch.

## 4 OPTIMIZATION

Our objective function comprises terms computed in 2D (on the images plane), and in 3D (on the triangles).

### 4.1 RENDERING LOSSES

Following previous work (Kerbl et al., 2023; Huang et al., 2024), we compute the SSIM between the rendered and input images, $\mathcal{L}_{ssim}$, and apply a normal consistency loss (Huang et al., 2024), $\mathcal{L}_{norm}$,

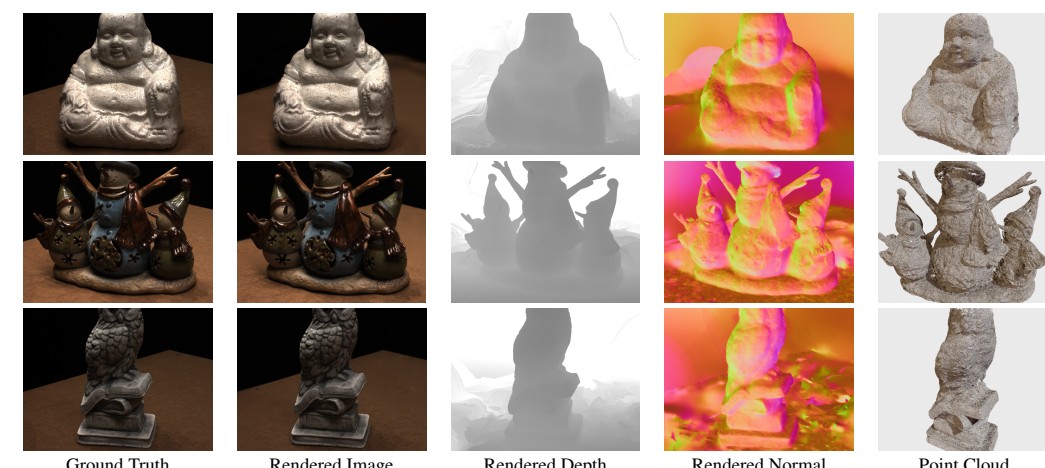

| Ground Truth | Rendered Image | Rendered Depth | Rendered Normal | Point Cloud |

Figure 7: Qualitative results on the DTU dataset (Aanæs et al., 2016). RTS estimates high-quality geometry, maintaining thin structures, such as the carrot noses and branches (middle row) and the book edges (bottom row), while effectively modeling smooth surfaces, such as the Buddha (top row).

to help locally align the triangles with the rendered surface. Please see Section S.3 the supplement for further details.

From the unsupervised depth estimation literature (Chang et al., 2022; Godard et al., 2017; Mahjourian et al., 2018), we adopt a smoothness term on the rendered depth map conditioned on the gradient of the input image. This penalizes large gradients in the rendered depth map where we have small gradients in the input image,

$$\mathcal{L}_{smooth} = \frac{1}{N}\sum_{i,j}||\partial_x D_{i,j}||e^{||\partial_x I_{i,j}||} + ||\partial_y D_{i,j}||e^{||\partial_y I_{i,j}||} \tag{3}$$

### 4.2 SCENE LOSS

To encourage connectivity between primitives, we penalize the mean of the $L2$ distance between the connected vertex pairs of neighboring triangle edges,

$$\mathcal{L}_{conn} = \sum_{a\in\Omega}\frac{1}{2}(||V_a^1 - V_b^1||_2 + ||V_a^2 - V_b^2||_2) + (1 - \widehat{n}_a^T\widehat{n}_b) \tag{4}$$

where $V_a^j$ and $V_b^j$ are the $j^{th}$ vertex pair of connected edges $a$ and $b$, respectively. $\widehat{n}_a$ and $\widehat{n}_b$ are the normals of the connecting triangles and $\Omega$ is the set of all triangles that intersect the current camera frustum. **The normal regularization is a soft penalty that encourages connected triangles to have similar normals to better align with the surfaces in the scene.** Applying this loss to invisible triangles without rendering losses leads to over-smoothing.

The final objective is a weighted summation of all terms:

$$\mathcal{L} = \omega_0\mathcal{L}_{ssim} + \omega_1\mathcal{L}_{norm} + \omega_2\mathcal{L}_{smooth} + \omega_3\mathcal{L}_{conn} \tag{5}$$

Table 1: Chamfer distance evaluation on scenes from DTU (Aanæs et al., 2016). Following previous literature, we average the accuracy and completeness Chamfer distances on the widely used evaluation set. Chamfer distances are measured in millimeters. The best results are in boldface and the second best are underlined.

| Method | \multicolumn{16}{c}{DTU} |
|---|---|
| | 24 | 37 | 40 | 55 | 63 | 65 | 69 | 83 | 97 | 105 | 106 | 110 | 114 | 118 | 122 | Mean (mm)↓ |
| 3DGS (Kerbl et al., 2023) | 2.14 | 1.53 | 2.08 | 1.68 | 3.49 | 2.21 | 1.43 | 2.07 | 2.22 | 1.75 | 1.79 | 2.55 | 1.53 | 1.52 | 1.50 | 1.96 |
| SuGaR (Guédon & Lepetit, 2024) | 1.47 | 1.33 | 1.13 | 0.61 | 2.25 | 1.71 | 1.15 | 1.63 | 1.62 | 1.07 | 0.79 | 2.45 | 0.98 | 0.88 | 0.79 | 1.33 |
| 2DGS (Huang et al., 2024) | 0.48 | 0.91 | 0.39 | 0.39 | 1.01 | 0.83 | 0.81 | 1.36 | 1.27 | 0.76 | 0.70 | 1.40 | 0.40 | 0.76 | 0.52 | 0.80 |
| GOF (Yu et al., 2024) | 0.50 | 0.82 | **0.37** | 0.37 | 1.12 | 0.74 | 0.73 | 1.18 | 1.29 | 0.68 | 0.77 | 0.90 | 0.42 | 0.66 | 0.49 | 0.74 |
| Gaussian Surfels (Dai et al., 2024) | 0.66 | 0.93 | 0.54 | 0.41 | 1.06 | 1.14 | 0.85 | 1.29 | 1.53 | 0.79 | 0.82 | 1.58 | 0.45 | 0.66 | 0.53 | 0.88 |
| PGSR (Chen et al., 2024a) | **0.36** | **0.57** | 0.38 | **0.33** | 0.78 | **0.58** | **0.50** | 1.08 | **0.63** | 0.59 | **0.46** | **0.54** | **0.30** | **0.38** | **0.34** | **0.52** |
| TriangleSplatting (Held et al., 2025a) | 0.98 | 1.07 | 1.07 | 0.51 | 1.67 | 1.44 | 1.17 | 1.32 | 1.75 | 0.98 | 0.96 | 1.11 | 0.56 | 0.93 | 0.72 | 1.06 |
| **RTS** | 0.42 | 0.61 | 0.74 | 0.39 | **0.53** | 0.86 | 0.70 | **0.84** | 0.72 | **0.37** | 0.68 | 0.87 | 0.34 | 0.58 | 0.44 | 0.61 |

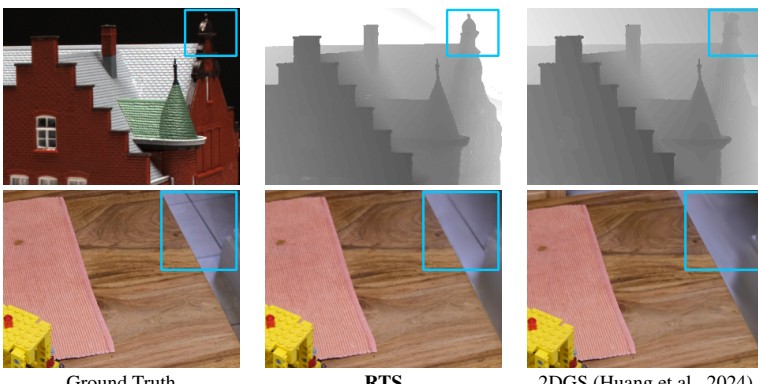



Ground Truth        **RTS**        2DGS (Huang et al., 2024)



Figure 8: Qualitative comparison between RTS and 2DGS (Huang et al., 2024) on; **Top** - *scan024* from the DTU dataset (Aanæs et al., 2016), **Bottom** - *kitchen* from the mip-NeRF 360 dataset (Barron et al., 2022). RTS is substantially more precise at estimating the geometry at discontinuities and rendering fine details, shown in the areas marked by the blue rectangles.

## 5 EXPERIMENTS

### 5.1 IMPLEMENTATION DETAILS

We implement the majority of our RTS framework in Python using PyTorch (Paszke et al., 2019). For rasterization, we develop custom CUDA kernels for both the forward and backward pass. We run all our experiments on a single NVIDIA RTX A6000. For our loss weights, we choose $\omega = [1.0 \quad 0.05 \quad 50.0 \quad 1000.0]$ empirically.

Following state-of-the-art Multi-View Stereo methods (Yao et al., 2018; Yang et al., 2022; Mi et al., 2022), we directly generate a 3D point cloud for geometric evaluation from the rendered depth maps *without performing any TSDF fusion*. To generate each point cloud, we use simple heuristic filtering on each depth map, similar to the post-processing presented in GBiNet (Mi et al., 2022). For each depth map, we measure the reprojection error of the depth values at every pixel using neighboring views and filter pixels based on this error. All depth estimates with a low reprojection error are back-projected to 3D points, forming the combined point cloud.

### 5.2 EVALUATION

We test our framework on the DTU dataset (Aanæs et al., 2016), an indoor dataset that contains images of 124 scenes taken from a camera mounted on an industrial robot arm. All scenes share the same camera trajectories, with ground-truth point clouds captured via structured light.

We evaluate our new approach on the DTU dataset and record the Chamfer distance in Table 1. We show competitive results alongside the leading state-of-the-art planar GS methods. Across the test set, RTS is the most geometrically accurate in several scenes and second most overall. While

Table 2: Novel View Synthesis on all scenes from the mip-NeRF 360 dataset (Barron et al., 2022).

| Method | Outdoor | | | Indoor | | |
|---|---|---|---|---|---|---|
| | PSNR↑ | SSIM↑ | LPIPS↓ | PSNR↑ | SSIM↑ | LPIPS↓ |
| NeRF (Mildenhall et al., 2020) | 21.46 | 0.458 | 0.515 | 26.84 | 0.790 | 0.370 |
| Deep Blending (Hedman et al., 2018) | 21.54 | 0.524 | 0.364 | 26.40 | 0.844 | 0.261 |
| Instant NGP (Müller et al., 2022) | 22.90 | 0.566 | 0.371 | 29.15 | 0.880 | 0.216 |
| MipNeRF360 (Barron et al., 2022) | 24.47 | 0.691 | 0.283 | **31.72** | 0.917 | 0.180 |
| SuGaR (Guédon & Lepetit, 2024) | 22.93 | 0.629 | 0.356 | 29.43 | 0.906 | 0.225 |
| 3DGS (Kerbl et al., 2023) | 24.64 | 0.731 | 0.234 | 30.41 | 0.920 | 0.189 |
| 2DGS (Huang et al., 2024) | 24.34 | 0.717 | 0.246 | 30.40 | 0.916 | 0.195 |
| GOF (Yu et al., 2024) | **24.82** | 0.750 | **0.202** | 30.79 | 0.924 | 0.184 |
| PGSR (Chen et al., 2024a) | 24.76 | **0.752** | 0.203 | 30.36 | **0.934** | 0.147 |
| 3DCS (Held et al., 2025b) | 24.07 | 0.700 | 0.238 | 31.33 | 0.927 | 0.166 |
| TriangleSplatting (Held et al., 2025a) | 24.27 | 0.722 | 0.217 | 30.80 | 0.928 | 0.160 |
| **RTS** | 21.41 | 0.657 | 0.349 | 30.28 | 0.921 | **0.130** |

Table 3: Ablation study on the contribution of the 3D loss terms using the entire DTU evaluation set. Here, $\lambda_c$ is the loss weight for the connectivity term, $L_{conn}$, and $\lambda_s$ is the loss weight for the depth smoothness term $L_{smooth}$. The best results are in boldface and the worst are underlined.

| Method | Acc.(mm) ↓ | Comp.(mm) ↓ | SSIM ↑ | PSNR ↑ | LPIPS ↓ | Primitives (K) ↓ |
|---|---|---|---|---|---|---|
| w/o $\mathcal{L}_{smooth}$ & $\mathcal{L}_{conn}$ | 0.66 | 0.68 | 0.912 | 30.58 | 0.227 | **218** |
| w/o $\mathcal{L}_{smooth}$ | 0.65 | 0.71 | **0.921** | **32.13** | **0.212** | 244 |
| w/o $\mathcal{L}_{conn}$ | 0.67 | 0.69 | 0.908 | 30.15 | 0.231 | 224 |
| full ($\lambda_c = 10.0$, $\lambda_s = 0.8$) | 0.64 | 0.70 | 0.918 | 31.68 | 0.213 | 249 |
| full ($\lambda_c = 300.0$, $\lambda_s = 20.0$) | 0.61 | 0.63 | 0.910 | 30.87 | 0.232 | 244 |
| full ($\lambda_c = 1000.0$, $\lambda_s = 50.0$) | **0.59** | **0.62** | 0.909 | 30.57 | 0.232 | 297 |

PGSR demonstrates impressive reconstruction results, the algorithm utilizes a full suite of multi-view objective functions that significantly improve the geometric reconstruction quality. We provide qualitative results on the DTU dataset in Fig. 7, showing visualizations of the rendered images, depth maps, normal maps, and final point clouds for three scenes. In Fig. 8 (*top*), we provide a comparison of depth map renderings between RTS and 2DGS (Huang et al., 2024). RTS is able to reconstruct fine details on the surfaces of objects that are typically blurred with Gaussian representations.

We show additional results on the mip-NeRF 360 dataset (Barron et al., 2022). Following the protocol specified by Barron et al. (2022), we separate the images in each scene, taking every eighth image as a test image and training on the remaining. As standard evaluation, we report PSNR, SSIM, and LPIPS (Zhang et al., 2018) metrics. We show results on all indoor and outdoor scenes in Table 2, as well as per-scene results in Table S.2. For indoor scenes, RTS shows results on par with the state-of-the-art methods, having the leading LPIPS score, the fourth overall SSIM score, and a highly competitive PSNR score among all listed methods. On two outdoor scenes (treehill and flowers), RTS is limited in reconstructing extremely distant background foliage, impacting the overall metrics (please see Fig. S.5 for qualitative results on these scenes). In Fig. 8 (*bottom*), we provide a comparison of novel view synthesis between RTS and 2DGS. RTS is able to render fine textures in low visibility regions in scenes as opposed to blurring with Gaussian representations.

## 5.3 ABLATIONS

We show an ablation evaluating the contributions of proposed loss terms in Table 3. Removing the soft connectivity leads to a decrease in overall accuracy of the output models, while removing the depth supervision negatively affects the completeness. The two supervision signals complement each other, and we show that increasing the loss weights allows for tuning the framework for either better novel view synthesis or geometry. Additionally, the results corresponding to the changes in magnitude of the loss weights demonstrates the stability of RTS to changes in hyper-parameters.

To evaluate geometry, all previous planar GS algorithms utilize a GT foreground-background segmentation mask when generating the final models. Using this mask to generate the final models for evaluation removes the effects of floaters and inaccurate estimation near the surfaces being evaluated. To portray a more grounded evaluation of geometry, we provide an ablation study in Table S.3 in which we compute Chamfer distance of models without the use of any GT masks.

## 6 LIMITATIONS & CONCLUSIONS

In this work, we introduce a new scene representation, namely Radiant Triangle Soup (RTS). To the best of our knowledge, we are the first to introduce explicit 3D forces between primitives in a splatting framework, helping to coordinate the positioning of primitives to directly form surfaces. Modifying the weights of these forces allows for tuning between 3D reconstruction quality and novel-view synthesis quality. The main limitation of our current algorithm is its inability to extract watertight meshes. Furthermore, due to the periodic nearest-neighbors search, there is a minor increase in run-time proportional to the number of primitives. Please see Section S.2 for more details.

The introduction of Triangle Soup as the underlying representation for Radiance Fields is amenable to future work in surface optimization. We plan to extend RTS with modified primitive connectivity strategies and perform optimization over watertight meshes.

## 7 REPRODUCIBILITY

In order to ensure reproducibility, we supplement the description of our method in Section 3 with broad implementation details in Section 5.1, as well as a complete account of hyper-parameter values used in our experiments in Section S.2 of the Supplemental Material. We will make our code publicly available to the research community, if the paper is accepted.

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

# SUPPLEMENTAL MATERIAL

Here we include additional material on background geometry used for computing barycentric coordinates, further implementation details, and additional results.

## S.1 BACKGROUND

In this section, we introduce the necessary background geometry used to parameterize a ray-triangle intersection in barycentric coordinates. The *barycentric coordinates* of a point on a triangle $\lambda = [\lambda^0, \lambda^1, \lambda^2]^T$, s.t. $\sum_{j=1}^3 \lambda^j = 1$, provide a means for expressing the coordinates of the point as a linear combination of the coordinates of the three vertices, $[V^0, V^1, V^2]$. This parameterization is important for graphics applications to be able to efficiently rasterize triangles onto screen space and interpolate the color of a pixel from each vertex.

In rendering, we compute the ray-triangle intersections using,

$$P = C + (\widehat{r}d) \tag{S.1}$$

where $C \in \mathbb{R}^3$ is the camera center, $\widehat{r} \in \mathbb{R}^3$ is the unit vector for the ray through pixel $i$, and $d \in \mathbb{R}$ is the depth along the ray from the camera center to the intersection point, computed as follows,

$$d = \frac{\widehat{n} \cdot \overrightarrow{CB}}{\widehat{n} \cdot \widehat{r}} \tag{S.2}$$

where $B \in \mathbb{R}^3$ is the barycenter of the triangle and $\widehat{n} \in \mathbb{R}^3$ is its normal.

We compute the barycentric coordinates for point $P$ using the triple products between the normal, a triangle edge, and the vector from each vertex to the point,

$$\lambda = \frac{1}{\widehat{n} \cdot (\overrightarrow{V_0V_1} \times \overrightarrow{V_0V_2})} \begin{bmatrix} \widehat{n} \cdot (\overrightarrow{V_1V_2} \times \overrightarrow{V_1P}) \\ \widehat{n} \cdot (\overrightarrow{V_2V_0} \times \overrightarrow{V_2P}) \\ \widehat{n} \cdot (\overrightarrow{V_0V_1} \times \overrightarrow{V_0P}) \end{bmatrix} \tag{S.3}$$

Intuitively, the contribution of each vertex is proportional to the area of the sub-triangle formed by the intersection point $P$ and the other two vertices of the triangle. This weight becomes larger as $P$ approaches the vertex.

## S.2 IMPLEMENTATION DETAILS

In this section, we describe our experimental setting and optimization parameters in detail. To begin optimization, similar to previous work (Huang et al., 2024; Chen et al., 2024a), all geometric supervision is disabled, with optimization only being guided initially by the SSIM loss $\mathcal{L}_{ssim}$. We enable the normal consistency loss $\mathcal{L}_{norm}$ at iteration $7,000$ and enable both the smoothness loss $\mathcal{L}_{smooth}$ and connectivity loss $\mathcal{L}_{conn}$ at iteration $10,000$, both chosen empirically. All triangles start with an initial opacity ($\alpha$) value set to $0.1$, with opacity for all primitives being reset every $3,000$ iterations. We run optimization on the scenes from the DTU and Mip-NeRF 360 datasets for $25,000$ and $30,000$ iterations, respectively. Densification and pruning is run every $250$ iterations starting after iteration $2,000$. The maximum incenter gradient threshold for densification is set to $7.5e^{-5}$ in all experiments.

The learning rates for each respective parameter are set as follows:

- Spherical Harmonics ($\Delta$): $2.5e^{-3}$
- Opacity ($\alpha$): $5e^{-2}$
- Incenter ($\mu$): $\begin{bmatrix} 1.5e^{-4}, & 2e^{-6} \end{bmatrix}$
- Rotation ($R$): $1e^{-3}$
- Scale ($s$): $4e^{-3}$
- Diffuse Scalar ($\sigma$): $1e^{-3}$

Table S.1: **Run-time comparison between baseline works and RTS on the DTU dataset (Aanæs et al., 2016).**

| Method | FPS | Run-Times |
|---|---|---|
| 2DGS (Huang et al., 2024) | 1400 | 0.41 hr |
| PGSR (Chen et al., 2024a) | 1200 | 1.05 hr |
| **RTS** | 400 | 1.50 hr |

where the incenter learning rate follows an exponential decay scheduler starting with $1.5e^{-4}$ and ending with $2e^{-6}$ conditioned on the total number of iterations. We also tested a linear decay scheduler leading to similar results.

**The initial diffuse scalar, $\sigma_0$, is inversely proportional to the mean distance, $d$, between each primitive and the three nearest neighboring primitives,**

$$\sigma_0 = \frac{log\left(\left(\frac{\alpha_0}{\gamma}\right) - 1\right)}{d} \tag{S.4}$$

**where $\alpha_0$ is the initial opacity value and $\gamma$ corresponds to the minimum $\alpha w_\sigma$ value necessary for a triangle to be rasterized in the forward pass of the network.**

**Edge Connectivity Overhead**    To compute neighboring edge connections, we construct a single KD-Tree containing the midpoints of all triangle edges which is then queried for each edge once. The connected edge indices are the only structure that is stored during optimization. Since the parameters of the primitives are modified during optimization, we need to recompute the KD-Tree and connected edge indices every 250 iterations (aligned with Adaptive Density Control), which ultimately leads to a minimal overhead for moderate size scenes. To be concrete, building the KD-Tree and computing the neighboring indices takes on average 4 seconds for around 300,000 primitives, which is more than the average number of primitives needed to reconstruct the scenes for the DTU dataset. Since this operation only happens every 250 iterations starting after iteration 10,000 (when the connectivity loss is activated), the overhead of this operation only adds roughly 5 minutes to the optimization process, which is about a $6\%$ increase in runtime. Reconstructing larger scenes, or more precisely, scenes that require more primitives, will naturally demand a larger overhead.

**Run-Time**    The run-times for our approach are roughly $1.5$ hours on DTU and $4.5$ hours on Mip-NeRF 360, with scenes from DTU and scenes from Mip-NeRF 360 having on average $249,941$ and $1,275,985$ primitives, respectively, using $0.5$ resolution for the DTU dataset and $0.25$ resolution for the Mip-NeRF 360 dataset. **Please see Table S.1 for a comparison with baselines**.

**Related Work Extension**    **In their concurrent work, TriangleSplatting, Held et al. (2025a) also propose a shift towards using triangle primitives as the explicit representation for inference-time optimization of a scene. In contrast to our work, each triangle in their approach is parameterized by a set of 3D points, similar to their previous work in 3D Convex Splatting (3DCS) (Vogel et al., 2015). The primitives in our work are parameterized by an incenter, scalar offsets to each vertex, and a rotation quaternion. Parameterizing the primitives in these separate terms allows for a more selective propagation of gradients with different learning rates, which in our experimentation, showed more favorable convergence toward a higher quality reconstruction. The activation function used in TriangleSplatting also differs from ours, as they use the normalized ReLU of the Signed Distance Function (SDF) for each triangle, whereas we apply a Sigmoid function to the SDF for each triangle. This is similar to the formulation used in 3DCS, however, we compute the exact SDF, compared to the approximation used in 3DCS. The primitives in our approach are also more expressive than that of TriangleSplatting. Each vertex encodes a separate color instead of using a single color for an entire triangle. As mentioned in the main paper, a key differentiator between the two works is the optimization criteria, in which our approach provides an avenue for primitive-to-primitive interactions through our soft connectivity forces. As shown in Table 1, RTS shows a major increase in geometric accuracy compared to TriangleSplatting. Both works have similar motivations, in which using triangles**

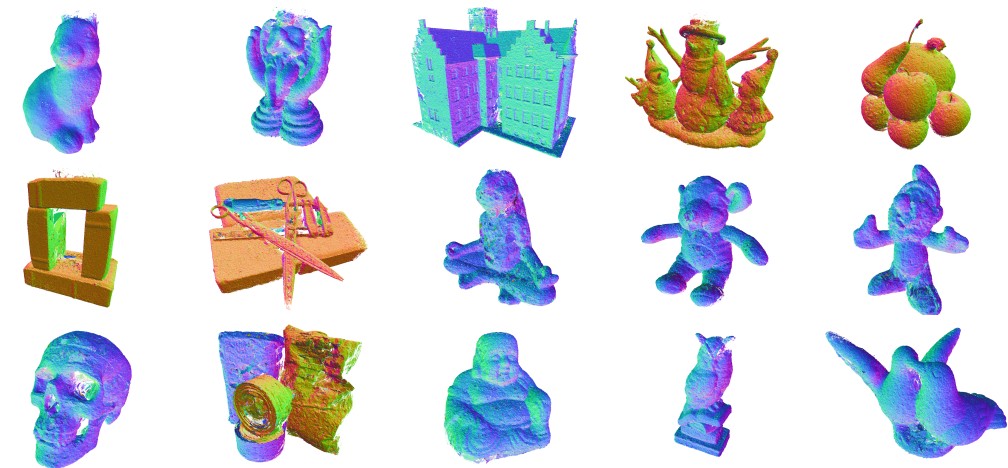

Figure S.1: **Output point cloud visualization of RTS for all scenes from the evaluation set from the DTU dataset(Aanæs et al., 2016). The points are colored according to their normals.**

Table S.2: Novel View Synthesis per-scene on all scenes from the mip-NeRF 360 dataset (Barron et al., 2022).

| Scene | SSIM | PSNR | LPIPS |
|---|---|---|---|
| room | 0.919 | 31.25 | 0.156 |
| counter | 0.900 | 28.27 | 0.139 |
| kitchen | 0.936 | 30.69 | 0.083 |
| bonsai | 0.930 | 30.93 | 0.142 |
| bicycle | 0.637 | 20.31 | 0.365 |
| flowers | 0.617 | 19.99 | 0.351 |
| garden | 0.742 | 24.89 | 0.250 |
| stump | 0.658 | 23.25 | 0.410 |
| treehill | 0.631 | 18.64 | 0.371 |

**as the primitive in an alpha-blending optimization framework provides a direct route to the estimation of a mesh with high-quality appearance that can directly be rendered in novel views.**

## S.3  LOSS FUNCTIONS

For completeness, we define the loss terms used in this paper that were introduced in previous work (Kerbl et al., 2023; Huang et al., 2024). The SSIM loss is computed as follows:

$$\mathcal{L}_{ssim} = \frac{1}{N} \sum_{i,j} (1-\gamma)L_1 + \gamma L_{D\text{-}SSIM} \tag{S.5}$$

The normal consistency loss is computed as follows:

$$\mathcal{L}_{norm} = \frac{1}{N} \sum_{i,j} (1 - \widehat{n}^T \widehat{n}_d) \tag{S.6}$$

This encourages alignment between the rendered normal $\widehat{n}$ with the normal computed from the rendered depth map $\widehat{n}_d$,

## S.4  ADDITIONAL EVALUATIONS

In Table S.2, we show the per-scene novel view synthesis evaluation for RTS on the mip-NeRF 360 dataset (Barron et al., 2022) using the standard metrics of PSNR, SSIM, and LPIPS. As von Lützow & Nießner (2025) note, using primitives with explicit boundaries can begin to introduce hard edges

Table S.3: Chamfer distance evaluation on scenes from DTU (Aanæs et al., 2016). Following previous literature, we average the accuracy and completeness Chamfer distances on the widely used evaluation set. Chamfer distances are measured in millimeters. **Top**: Chamfer distance evaluation using the GT mask. **Bottom**: Chamfer distance evaluation *without* using the GT mask.

| Method | DTU | | | | | | | | | | | | | | | Mean (mm)↓ |
|---|---|---|---|---|---|---|---|---|---|---|---|---|---|---|---|---|
| | 24 | 37 | 40 | 55 | 63 | 65 | 69 | 83 | 97 | 105 | 106 | 110 | 114 | 118 | 122 | |
| 2DGS (Huang et al., 2024) | 0.48 | 0.91 | 0.39 | 0.39 | 1.01 | 0.83 | 0.81 | 1.36 | 1.27 | 0.76 | 0.70 | 1.40 | 0.40 | 0.76 | 0.52 | 0.80 |
| PGSR (Chen et al., 2024a) | **0.36** | **0.57** | 0.38 | **0.33** | 0.78 | **0.58** | **0.50** | 1.08 | **0.63** | 0.59 | **0.46** | **0.54** | **0.30** | **0.38** | **0.34** | **0.52** |
| **RTS** | 0.42 | 0.61 | 0.74 | 0.39 | **0.53** | 0.86 | 0.70 | **0.84** | 0.72 | **0.37** | 0.68 | 0.87 | 0.34 | 0.58 | 0.44 | 0.61 |
| 2DGS (Huang et al., 2024) [no mask] | 1.22 | 1.69 | **0.88** | **0.43** | 0.96 | 0.77 | 0.85 | 1.23 | 1.87 | 1.69 | 0.91 | 2.01 | 0.82 | 0.82 | 1.07 | 1.15 |
| PGSR (Chen et al., 2024a) [no mask] | 1.10 | 1.59 | 0.97 | 0.45 | 1.98 | **0.64** | **0.59** | 1.70 | 1.69 | 1.57 | **0.74** | **0.63** | 0.44 | **0.71** | **0.74** | 1.04 |
| **RTS** [no mask] | **0.52** | **0.92** | 0.94 | 0.55 | **0.91** | 1.01 | 0.74 | **1.09** | **1.05** | 0.67 | 0.88 | 1.07 | **0.42** | 0.79 | 0.80 | **0.82** |

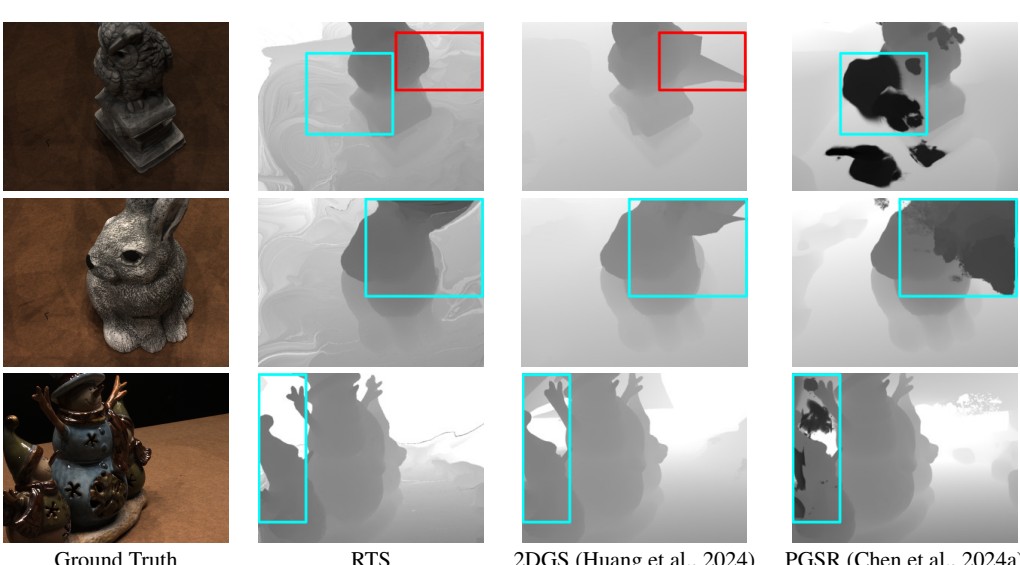

Ground Truth | RTS | 2DGS (Huang et al., 2024) | PGSR (Chen et al., 2024a)

Figure S.2: **Qualitative results between RTS and two baselines on the DTU dataset (Aanæs et al., 2016). RTS is much more effective at removing floaters. This is especially helpful in extreme viewpoints with low camera overlap.**

in regions with poor visibility while smoother primitives degrade more gracefully. While the diffuse boundaries of our triangles help prevent much of this behavior, its effects are noticeable in some of the reconstructed images, especially in outdoor scenes with distant background foliage. See Fig. S.5

As mentioned in Section 5.3, we compare the Chamfer distances of RTS and two competitive GS algorithms in Table S.3 with and without the use of the GT segmentation masks. While the distances increase for all methods, this experiment demonstrates how RTS is more effective at floater removal and background modeling.

**Qualitative comparisons to competitive baselines on the DTU dataset (Aanæs et al., 2016) are shown in Fig. S.2, and on the mip-NeRF 360 dataset (Barron et al., 2022) are shown in Fig. S.3.** Additionally, we provide renderings of novel views on challenging indoor and outdoor scenes with fine details and non-Lambertian surfaces. Output renderings and 3D models for all scenes will be made publicly available to the research community if the paper is accepted. We show qualitative geometric reconstruction results on all scenes from the DTU dataset in Fig. S.1 and novel view synthesis results on all indoor and outdoor scenes from the Mip-NeRF 360 datasets in Fig. S.4 and Fig. S.5, respectively.

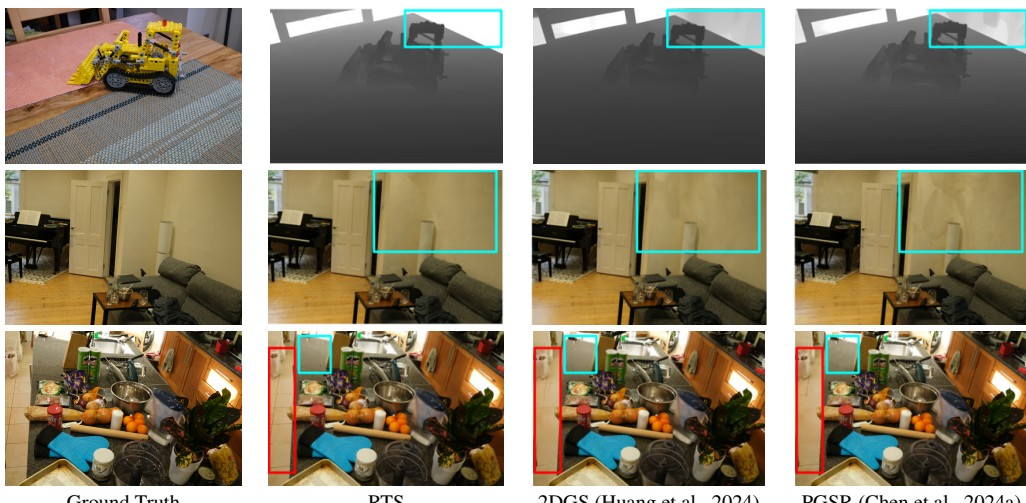


Ground Truth      RTS      2DGS (Huang et al., 2024)      PGSR (Chen et al., 2024a)


Figure S.3: **Qualitative results between RTS and two baselines on the mip-NeRF 360 dataset (Barron et al., 2022). RTS can better represent texture-less areas and recover sharp details from surfaces seen in few views.**

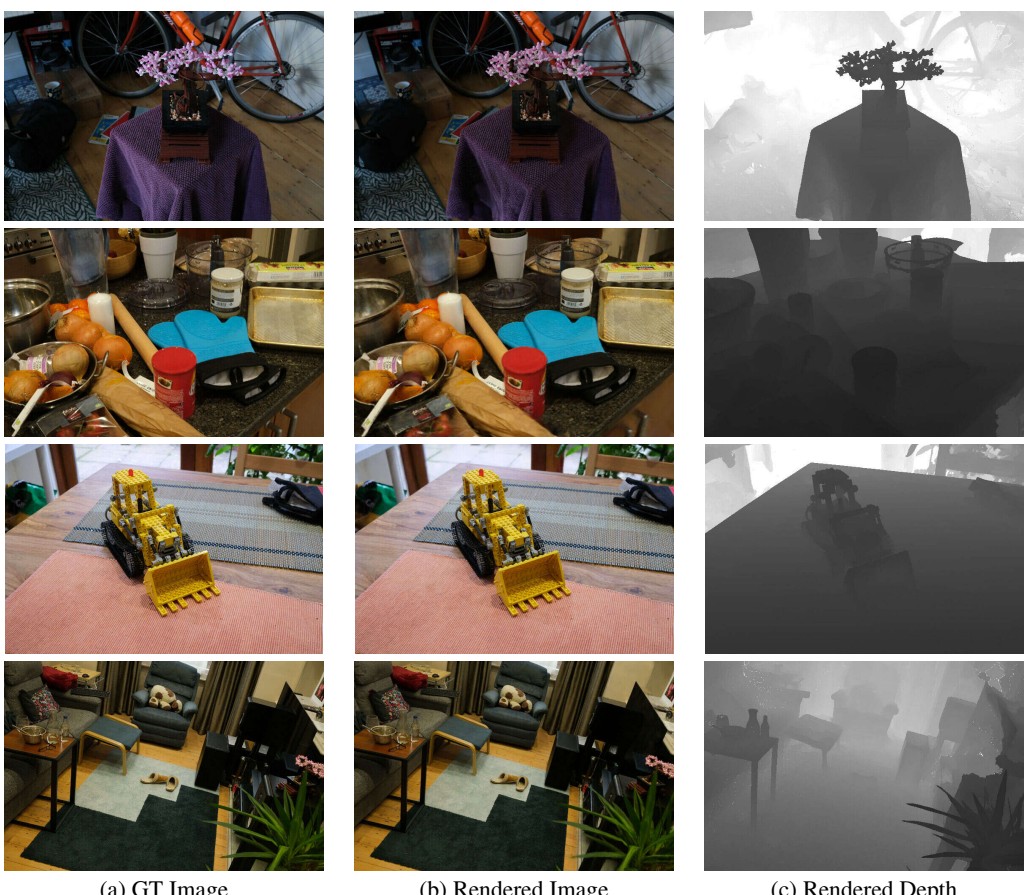


(a) GT Image      (b) Rendered Image      (c) Rendered Depth


Figure S.4: Visualizations of all indoor scenes, *(top-down) [bonsai, counter, kitchen, room]*, from the mip-NeRF 360 dataset (Barron et al., 2022).

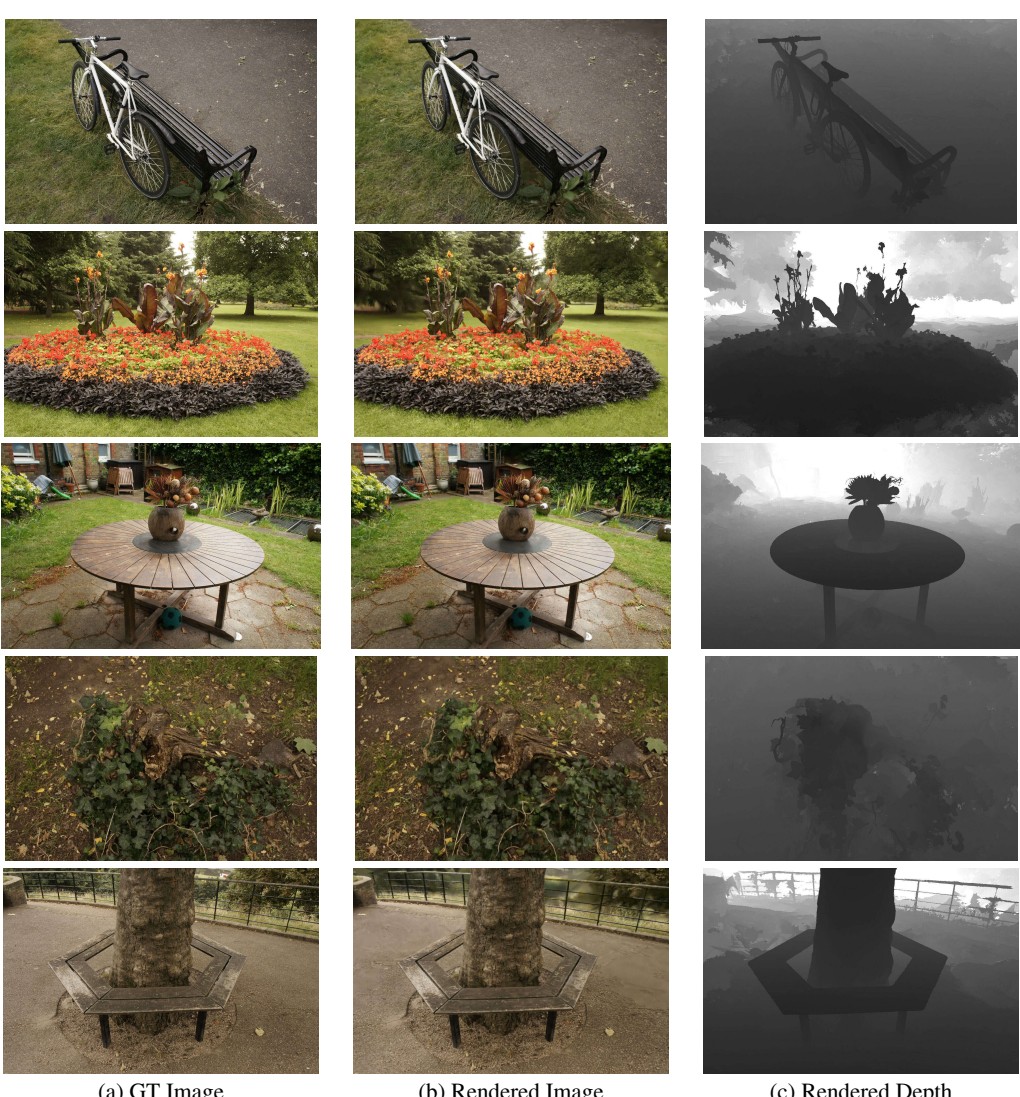

(a) GT Image          (b) Rendered Image          (c) Rendered Depth

Figure S.5: Visualizations of all outdoor scenes, *(top-down) [bicycle, flowers, garden, stump, treehill]*, from the Mip-NeRF 360 dataset (Barron et al., 2022). In the top-right section of the rendered image of treehill, RTS imprecisely approximates the appearance of the background foliage, while GS-based algorithms typically blur this region.

