# OpenReview forum: "Radiant Triangle Soup with Soft Connectivity Forces for 3D Reconstruction and Novel View Synthesis"
_ICLR.cc/2026/Conference — Submitted to ICLR 2026_

### Official Review · Reviewer_wzMp · 2025-10-26

**Soundness:** 3
**Presentation:** 3
**Contribution:** 2
**Rating:** 2
**Confidence:** 4

**Summary:**

This paper introduces a new idea of using triangles as the primitive to replace 3D Gaussians for the splatting. This method adopts a soft triangle representation to allow gradients to propagate to triangle positions and also considers the connectivity of triangles. The method is evaluated on the DTU dataset for surface reconstruction and the Mip-NeRF-360 dataset for novel view synthesis. The results demonstrate better geometry quality than Triangle splatting on the DTU dataset but worse rendering quality than triangle splatting on the novel view synthesis.

**Strengths:**

The idea of associating different triangles seems to be interesting and improved performance of geometry reconstruction is demonstrated.

**Weaknesses:**

1. The idea of making rasterization of triangles differentiable has already been studied in previous differentiable renderers like SoftRasterizer, and so on.
2. Why learning such connectivity is useful is not discussed clearly, which is the main contribution of the proposed method. I'm not sure in any cases, we need to connect the surface for a better rendering quality.
3. The search for connected triangles is costly. In the supp, the runtime for the DTU scene is 1.5h, which is much slower than baselines like GoF, PGSR, and so on.


In summary, the paper is not well motivated with new techniques. The proposed triangle connection seems to have an obvious weakness in terms of computation inefficiency. The results (for both geometry and NVS tasks) are not impressive enough either.

**Questions:**

I have no other questions than the weakness.

---

> ### Author Response · Authors · 2025-11-22
>
> We would like to thank the reviewer for their time and for their feedback and comments on our work.
>
> ## Weaknesses
> ### (1) "The idea of making rasterization of triangles differentiable has already been studied in previous differentiable renderers like SoftRasterizer, and so on..."
> This is correct; however, the approach that we present in this paper introduces triangle soup as the representation for an alpha-blending optimization algorithm. We would like to be very clear that our contributions do not include the formulation of a differentiable rasterization of triangles. If there are any specific instances in the paper that the reviewer feels are unclear on this point, we would be happy to address them directly and improve the clarity of our work.
>
> ### (2) "Why learning such connectivity is useful is not discussed clearly, which is the main contribution of the proposed method. I'm not sure in any cases, we need to connect the surface for a better rendering quality..."
>
> We feel the comment from Question #1 of Reviewer QKuE partially addresses this comment, which we have copied below, but to first summarize and to provide a more direct response to the concern:
>
> The motivation for using planar surfels in alpha-blending optimization (RTS, 2DGS, PGSR, etc.) is to utilize a primitive that better represents the surfaces in scenes and to reduce floating artifacts that tend to arise from the use of volumetric primitives, leading to more accurate 3D models (in both geometry and appearance).
>
> Optimizing for connectivity is directed at improving the geometry of the representation, which we observe in the ablation study in Table 3. Furthermore, our representation and connectivity loss significantly increases floater removal (please see the response to Weakness 1a of Reviewer TvUE for more details) which leads to much higher quality geometry without the use of ground truth foreground-background segmentation masks.
>
> Response to Reviewer QKuE:
> *It is important to first think about the desired output of a 3D reconstruction algorithm and the motivations behind Radiance Field optimization. The ultimate goal of Multi-View Stereo (MVS) is to obtain a geometrically accurate model of the object/scene from a collection of images, usually taking the form of a watertight mesh. In practice, appearance modeling is deemed out of scope. Radiance Fields focus much more on the appearance modeling of a scene, adding the ability to render high quality images from novel views. The ideal product of the these methods would be a 3D asset in the form of a mesh that is geometrically accurate, yet also models non-Lambertian, complex appearance of a scene and can be rendered from new viewpoints.*
>
> *To address the question, the disadvantage of Gaussian kernels (especially 3D kernels) being used as the underlying representation is the fact that the scene needs to be converted (usually through post-processing the rendered depth maps via TSDF-Fusion or a similar algorithm) into a triangle mesh in order to obtain a 3D model and to evaluate geometry. This removes any capability of modeling appearance, as the 3D model being evaluated can no longer be rendered. This limitation does not exist if the representation is already using triangles as primitives.*
>
> *Furthermore, the primary objective of this direction of research would be to obtain a watertight mesh directly from the optimization (or as you point out, potentially from a feed-forward model) that is directly compatible with modern rendering engines, and that ALSO maintains the appearance modeling afforded to Radiance Field representations.*
>
> *This is the main motivation behind our paper. And as we mention in our conclusion, we plan to extend this framework, experimenting with different soft connectivity strategies, as well as working towards watertight mesh optimization.*
>
> ### (3) "The search for connected triangles is costly. In the supp, the runtime for the DTU scene is 1.5h, which is much slower than baselines like GoF, PGSR, and so on...."
> To be precise, the edge connectivity overhead only adds roughly 5 minutes to the overall optimization. You can find a much more detailed description in lines 779-790 in the supplemental material.
>
> To reiterate our response on run-time to Reviewer TvUE, run-time is highly dependent on the implementation of the algorithm and the hardware it is executed on. For example, the PGSR publication reports an average runtime of 2 hours for GoF on the DTU dataset compared to 0.31 hours from the GoF publication. Additionally, the code of our implementation is an entirely new approach with many custom CUDA kernels and is not yet fully optimized.
>
> ### (4) "The proposed triangle connection seems to have an obvious weakness in terms of computation inefficiency..."
> Please see the response to Weakness #3 above.

---

> > ### Comment · Reviewer_wzMp · 2025-11-28
> >
> > Thanks for your reply! This addresses some of my concerns, but I still have the following issues I want to discuss.
> >
> > 1. The relationship between the proposed method and differentiable rasterization is clear. I would like to ask the authors to include some citations to rasterization methods because Fig. 4 looks quite similar to Fig. 4 of SoftRasterizer with a similar formulation.
> > 2. For the motivation of connectivity, 1) I would like authors to show some visualization of how the proposed term improves the surface quality by formulating a concrete surface. All the current discussion is based on the assumption that connecting them would be better, but this is not well validated by experiments. For example, it pulls several relevant triangles together.
> > 2) The authors are encouraged to add these discussions to the paper to make the paper look more well-motivated.
> > 3. For the runtime, I think it would be better to list a table of the time consumption of each stage and compare it with the reported runtime or your re-implemented runtime of baselines using the same running environment. Some discussions are needed here to clarify the differences.

---

> > > ### Author Response · Authors · 2025-12-03
> > >
> > > ### (1) "The relationship between the proposed method and differentiable rasterization is clear. I would like to ask the authors to include some citations to rasterization methods because Fig. 4 looks quite similar to Fig. 4 of SoftRasterizer with a similar formulation..."
> > > The parameterization for our work is based on the formulation presented by Held et al. in 3D Convex Splatting. While the intuition is similar to that of SoftRasterizer, the sigma parameter in our work, much like in 3DCS, is optimized for each primitive. Since their formulation is the most similar, we cite 3DCS in this section of the paper.
> > >
> > > ### (2) "For the motivation of connectivity, 1) I would like authors to show some visualization of how the proposed term improves the surface quality by formulating a concrete surface. All the current discussion is based on the assumption that connecting them would be better, but this is not well validated by experiments. For example, it pulls several relevant triangles together...."
> > > For experimental validation, we provide an ablation study removing the connectivity loss and increasing the weights in the total loss function in Table 3. We show that without the connectivity loss, the average Chamfer distance is 0.67 mm. With the connectivity loss, the Chamfer distance drops to 0.61 mm. Additionally, we provide the results for the official DTU evaluation in Table S.2, removing the curated segmentation mask. Without the segmentation mask, the impact of outliers propagates to the geometric evaluation. As a result of our parameterization and our connectivity loss, the final models produced by RTS have very few floaters, as evident by the robustness to the removal of the masking. The best media representation to present these effects visually is a video, which we plan to release along with the code for our implementation.
> > >
> > > ### (3) "For the runtime, I think it would be better to list a table of the time consumption of each stage and compare it with the reported runtime or your re-implemented runtime of baselines using the same running environment. Some discussions are needed here to clarify the differences."
> > > We have provided run-time metrics in Table S.1 in the updated version of the paper.

---

### Official Review · Reviewer_QKuE · 2025-10-27

**Soundness:** 3
**Presentation:** 3
**Contribution:** 3
**Rating:** 6
**Confidence:** 4

**Summary:**

This paper introduces Radiant Triangle Soup (RTS), a novel 3D scene representation for reconstruction and novel view synthesis that uses a set of translucent triangle primitives, with an additional contribution to enhancing a soft connectivity force between neighboring triangles, enabling explicit cross-primitive coordination during optimization.

**Strengths:**

1. The paper is well written and easy to follow.
2. RTS strikes an impressive balance between geometric fidelity and color expressiveness, offering a promising primitive for neural rendering and 3D reconstruction.
3. The soft connectivity force is interesting and enables direct information exchange among primitives, and it may be a unique characteristic of the triangular representation.
4. Rich implementation details are documented meticulously, ensuring the work can be reliably reproduced and extended.
5. I encourage the authors to continue exploring this type of representation, as it may offer better compatibility with modern rendering engines compared to 3D Gaussians.

**Weaknesses:**

I think the main weaknesses are as follows:
1. From the quantitative metrics, there is still a noticeable gap compared with SOTA methods, both in novel view synthesis and geometry reconstruction. The authors could consider introducing a regularization similar to that in 2DGS to further enhance the credibility of the experiments. Although this may make the approach more complex, it could lead to a fairer quantitative comparison.
2. If my understanding is correct, Triangle Splatting is a work that is quite similar in terms of representation. The authors should further clarify the core contributions of this paper, especially at the representational level.
3. Training efficiency and rendering FPS is not reported if I don't miss anything.

Point 2 is the main reason I gave a initial score of 6, and it strongly influences my inclination to adjust the score during rebuttal. I hope the authors can respond to this point carefully.

**Questions:**

1. How do the authors view the future development of the triangle-based representation? I believe that, given the current rapid progress in feed-forward reconstruction networks, the triangle-based form should also start exploring reconstruction approaches beyond optimization-based methods. On this basis, if the triangle representation evolves toward feed-forward reconstruction networks, would it have any advantages over 3D Gaussians?

2. From a deployment perspective, it seems that the triangle representation is significantly more compatible with modern rendering engines compared to 3D Gaussians. Is this true?

**Details Of Ethics Concerns:**

No.

---

> ### Author Response · Authors · 2025-11-22
>
> First and foremost, we would like to thank the reviewer for their time and for their feedback and comments on our work. We appreciate the detail provided in the response.
>
> ## Weaknesses
> ### (1a) "From the quantitative metrics, there is still a noticeable gap compared with SOTA methods, both in novel view synthesis and geometry reconstruction..."
>
> Addressing the comparison between our method and PGSR, please refer to our response to Weakness 1a of Reviewer TvUE, which we provide a copy of below:
>
> *PGSR is indeed related to our method, with many technical differences mainly due to the choice of primitive as well as the additional loss functions. Their framework applies heavy supervision across multiple views by rendering nearby images and enforcing reprojection criteria during optimization. These losses are indeed applicable to our framework; however, it was our intention to present this entirely new approach using triangle primitives, and a primitive-to-primitive loss, without implementing all of the algorithmic options afforded to methods dealing in Gaussian kernels, with the assumption that these algorithmic additions can be applied to RTS in future work.*
>
> *Along these lines, the intended baseline for the most direct comparison with our method is 2DGS. Employing the same SSIM and Normal Consistency losses, our use and parameterization of triangle primitives directly leads to improvements in geometric reconstruction of surfaces. This claim is supported by the first row in Table 3, in which we evaluate our approach using the same losses as 2DGS (without our additional losses), measuring 0.67 overall Chamfer distance on the DTU dataset compared to 0.80 for 2DGS (from Table 1).*
>
> *Furthermore, to remain consistent with previous literature, the evaluation in Table 1 utilizes a ground truth foreground-background segmentation mask. This mask is not provided by the DTU dataset and is not a part of the official benchmark evaluation, since its use significantly simplifies the reconstruction. In order to portray a more fair evaluation of reconstruction quality, we provide a comparison evaluating 2DGS, PGSR, and RTS with and without this ground truth mask (following the official DTU benchmark) in Table S.2 of the supplemental material. The Chamfer distance degradation is as follows: (i) 2DGS is 0.80 to 1.15; (ii) PGSR is 0.52 to 1.04; (iii) RTS is 0.61 to 0.82. We show that even though the reconstruction quality for all methods decreases, RTS is much more robust to the removal of the ground truth masks, as our method handles floating primitives much more effectively than the baseline methods.*
>
> *This motivates the need for direct losses on the primitives in 3D, whereas the multi-view losses applied in PGSR have little effect in regions of the scene with inconsistent geometry, which ultimately get filtered out by the ground truth masks.*
>
>
> ### (1b) "The authors could consider introducing a regularization similar to that in 2DGS to further enhance the credibility of the experiments..."
> Our losses consist of two of the three loss terms from 2DGS. We include both the SSIM and Normal Consistency losses from 2DGS, and choose to exclude the Distortion loss, since this term has very little effect on the quality of the reconstruction (both in appearance and in geometry).

---

> > ### Author Response · Authors · 2025-11-22
> >
> > ### (2) "Triangle Splatting is a work that is quite similar in terms of representation. The authors should further clarify the core contributions of this paper, especially at the representational level..."
> > Addressing the comparison between our method and TriangleSplatting, please refer to our response to Weakness 1a of Reviewer TvUE, which we provide a copy of below:
> >
> > *TriangleSplatting is an unpublished work that is concurrent with our own. As the reviewer mentions, we reference this paper as a part of related work and provide a quantitative comparison against this work in our evaluations, in which the authors kindly provided us with their geometric results.*
> >
> > *RTS shows a major increase in geometric accuracy compared to TriangleSplatting. As we mention in the section on related work, TriangleSplatting use translucent triangles as their primitive. Each triangle in their work is parameterized by a set of 3D points, similar to their previous work in 3D Convex Splatting. The primitives in our work are parameterized by an incenter, scalar offsets along the interior angle bisectors, and a rotation quaternion. Parameterizing the primitives in these separate terms allows for a more selective propagation of gradients with different learning rates, which in our experimentation, showed more favorable convergence toward a higher quality reconstruction. The activation function used in TriangleSplatting also differs from ours, as they use the normalized ReLU of the SDF for each triangle, whereas we apply a Sigmoid function to the SDF for each triangle. Both works have very similar motivations, in which using triangles as the primitive in an alpha-blending optimization framework provides a direct route to the estimation of a mesh with high-quality appearance that can directly be rendered in novel views.*
> >
> > *We agree with the reviewer, and will modify the text adding a more detailed description of the related method.*
> >
> > ### (3) "Training efficiency and rendering FPS is not reported if I don't miss anything..."
> > We do not report rendering FPS in this version of the paper since our emphasis is on reconstruction not fast rendering. We are confident that our representation can be rendered in real time. We do provide run-time and primitive count information in the supplemental material on lines 779-794, in which we report the run-time for our implementation is on average 1.5h and 4.5h with 249,941 and 1,275,985 primitives on the DTU and mip-NeRF 360 datasets, respectively.

---

> ### Author Response · Authors · 2025-11-22
>
> ## Questions
> ### (1) "How do the authors view the future development of the triangle-based representation? I believe that, given the current rapid progress in feed-forward reconstruction networks, the triangle-based form should also start exploring reconstruction approaches beyond optimization-based methods. On this basis, if the triangle representation evolves toward feed-forward reconstruction networks, would it have any advantages over 3D Gaussians?..."
> It is important to first think about the desired output of a 3D reconstruction algorithm and the motivations behind Radiance Field optimization. The ultimate goal of Multi-View Stereo (MVS) is to obtain a geometrically accurate model of the object/scene from a collection of images, usually taking the form of a watertight mesh. In practice, appearance modeling is deemed out of scope. On the other hand, Radiance Fields focus much more on the appearance modeling of a scene, adding the ability to render high quality images from novel views. The ideal product of these methods would be a 3D asset in the form of a mesh that is geometrically accurate, yet also models non-Lambertian, complex appearance of a scene and can be rendered from new viewpoints.
>
> To address the question, the disadvantage of Gaussian kernels (especially 3D kernels) being used as the underlying representation is the fact that the scene needs to be converted (usually through post-processing the rendered depth maps via TSDF-Fusion or a similar algorithm) into a triangle mesh in order to obtain a 3D model and to evaluate geometry. This removes any capability of modeling appearance, as the 3D model being evaluated can no longer be rendered. This limitation does not exist if the representation is already using triangles as primitives.
>
> Furthermore, the primary objective of this direction of research would be to obtain a watertight mesh directly from the optimization (or as you point out, potentially from a feed-forward model) that is directly compatible with modern rendering engines, and that ALSO maintains the appearance modeling afforded to Radiance Field representations.
>
> This is the main motivation behind our paper. And as we mention in our conclusion, we plan to extend this framework, experimenting with different soft connectivity strategies, as well as working towards watertight mesh optimization.
>
> ### (2) "From a deployment perspective, it seems that the triangle representation is significantly more compatible with modern rendering engines compared to 3D Gaussians. Is this true?..."
> This has been addressed by the above comments.

---

> > ### Comment · Reviewer_QKuE · 2025-11-26
> >
> > I appreciate the authors' detailed response. However, there are still some issues I hope the authors can clarify:
> >
> > 1. For evaluation, I think the authors' explanation is still rather weak. A good method should ensure that the benchmark is fair and then outperform (or partially outperform) other baselines to prove its effectiveness. If the baselines use other regularizations or annotations, either you should use them too, or all should be removed, just make it fair.
> >
> > 2. Regarding the comparison with Triangle Splatting, the authors' clarification has instead made me concerned about the novelty of the work, as the authors mainly emphasize the differences in parameterization and activation. Could you please clarify whether Triangle Splatting is a concurrent work or a prior work relative to this paper?
> >
> > 3. For efficiency, I do not quite understand the authors' clarification. The rendering FPS is very important, both for readers to choose between different schemes and for the improvement of subsequent work. I hope the authors can directly use a clear quantitative table for comparison.

---

> > > ### Author Response · Authors · 2025-11-27
> > >
> > > ### (1) "For evaluation, I think the authors' explanation is still rather weak. A good method should ensure that the benchmark is fair and then outperform (or partially outperform) other baselines to prove its effectiveness. If the baselines use other regularizations or annotations, either you should use them too, or all should be removed, just make it fair."
> > > As to not stray from the previous evaluation method used in the literature, we report the Chamfer distance on the DTU dataset using this supplemental foreground-background segmentation mask in Table 1 of the main paper. In addition, we provide metrics using the official evaluation of the benchmark in Table S.2. It is informative to include both for the sake of a complete evaluation. That being said, RTS performs well in both settings, outperforming every baseline aside from PGSR, with an overall Chamfer distance of 0.61 compared to 0.52 for PGSR. Additionally, RTS shows impressive robustness to the removal of ground truth segmentation, unlike the baseline works.
> > >
> > > We would be happy to provide additional statistics on these results, if the Reviewer has something else in mind.
> > >
> > > ### (2) "Regarding the comparison with Triangle Splatting, the authors' clarification has instead made me concerned about the novelty of the work, as the authors mainly emphasize the differences in parameterization and activation. Could you please clarify whether Triangle Splatting is a concurrent work or a prior work relative to this paper?"
> > > TriangleSplatting is a concurrent, unpublished work which was posted on arxiv at the time of submission to ICLR. Apparently, it has since been accepted to 3DV 2026, which will be held in March. We consider this work concurrent because no refereed version was available when we submitted our paper. For a more compact and direct description, the two methods differ in the following:
> > >
> > > - initialization
> > > - parameterization
> > > - diffuse activation
> > > - color representation per primitive
> > > - densification and pruning
> > > - objective functions
> > >
> > > We will modify the related works section to reflect these differences more clearly.
> > >
> > > ### (3) "For efficiency, I do not quite understand the authors' clarification. The rendering FPS is very important, both for readers to choose between different schemes and for the improvement of subsequent work. I hope the authors can directly use a clear quantitative table for comparison."
> > > We do not claim that we have made any contributions to the rendering aspects of the pipeline. RTS generates a set of triangles with spherical harmonics as color parameterization with a comparable number of primitives to baselines using ellipsoids. Rendering these triangles can be performed efficiently by standard graphics pipelines. For completeness, we will provide a table comparing the rendering frame rates between our implementation and the baselines in an updated version of the paper.

---

> > > > ### Comment · Reviewer_QKuE · 2025-11-27
> > > >
> > > > I appreciate the authors' reply. My concerns regarding the evaluation and the comparison with Triangle Splatting have been resolved. Regarding efficiency, I hope the authors can add the relevant table in the revision as soon as possible.
> > > >
> > > > I decided to keep my score for now, but may change depending on my entire review batch.

---

### Official Review · Reviewer_avbo · 2025-10-31

**Soundness:** 4
**Presentation:** 3
**Contribution:** 3
**Rating:** 8
**Confidence:** 3

**Summary:**

Scene reconstruction / analysis-by-synthesis method using triangles as a primitive compared to existing primitive reconstruction works like gaussian splats. Furthermore learns connectivity between primitives allowing you to get surfaces. This blurs the line between mesh based methods for which optimizing over strange topologies can be difficult, and more loose primitive methods like gaussian splats for which you don't get surface normals that make a lot of sense.

**Strengths:**

I really enjoyed reading the paper, it was well motivated, and is a really original and interesting idea. Lots of well designed qualitative results and a lot of quantitative results. Not quite SOTA but demonstrates competitive results.

**Weaknesses:**

I really want there to be an additional table to include losses used in all these different methods. My main concern is that this uses normal information during training while other methods like 3DGS don't require that information ahead of time, and while you ablated the other loss terms, you didn't ablate this term. I'm concerned that this limits the usage of this to synthetic or highly constrained setups where you have the normal.

**Questions:**

I'd love some more perspective to ease my concerns about normals being required.

---

> ### Author Response · Authors · 2025-11-22
>
> We would like to thank the reviewer for their time and for their feedback and comments on our work.
>
> ## Weaknesses
> ### (1) "I really want there to be an additional table to include losses used in all these different methods..."
> We agree. It would be a very useful contribution to the community to evaluate the effects of different objective functions isolated from the underlying representations and pruning/densification algorithms. However, we feel that this is better suited for a survey paper on the topic.
>
> ### (2) "My main concern is that this uses normal information during training while other methods like 3DGS don't require that information ahead of time, and while you ablated the other loss terms, you didn't ablate this term..."
> To clarify on the normal consistency loss included in this paper, we do not require any normal maps to be precomputed during optimization. The normal consistency loss is the exact same loss used in 2DGS. It is simply the regularization between the rendered normals (through alpha-blending) and the normals computed from the rendered depth maps. This is meant as a geometric consistency supervision, with the intention that the normal map computed from the rendered depth map should be consistent with the rendered normals directly from the primitives, encouraging primitive alignment with the rendered geometry. For completeness, when discussing the normal of the primitives, the eigenvector of the covariance matrix corresponding to the minimum eigenvalue is used as the normal of each splat in GS frameworks, while we directly use the normal of the triangle in our work.
>
> There are no ground-truth normal maps used in our approach. Both terms in the loss function are a direct product of the rasterization of the triangle soup.
>
> The reason we did not ablate these loss terms is because they are inherited from 2DGS. Our ablations were targeted at evaluating the contribution of the use and parameterization of triangle primitives, separate from the contribution of primitive-to-primitive loss terms that are introduced in this work.
>
> ### (3) "I'm concerned that this limits the usage of this to synthetic or highly constrained setups where you have the normal..."
> This concern is mostly answered in the response to Weakeness #2 above. To summarize, our approach is not limited to scenarios where normals exist for our training views, as these are not used in our pipeline. Both of the normal maps used in our normal consistency loss come directly from rendered maps of our scene representation.

---

### Official Review · Reviewer_TvUE · 2025-10-31

**Soundness:** 2
**Presentation:** 3
**Contribution:** 2
**Rating:** 2
**Confidence:** 4

**Summary:**

The paper proposes Radiant Triangle Soups as a new 3D representation that can be optimized via differentiable rendering and aims for high-quality 3D surface reconstruction and novel view synthesis at the same time. It leverages triangles as primitives, parameterized by the center, per-vertex Spherical Harmonic color coefficients, scales of vectors from the center to the vertices, a 3D rotation, a scalar opacity, and a diffuse strength, with the latter being necessary to obtain gradients for optimization.
Following the previous work of 3D Gaussian Splatting and follow-ups, the triangles are initialized using a point cloud obtained via SfM, rasterized in a differentiable fashion from training views, and optimized for reconstructing the ground truth images. Additionally to established losses like SSIM and depth smoothness, the authors propose a scene loss that softly encourages connectivity between neighboring triangles with aligned normals and rotations with the goal being emerging closed surfaces during optimization.
The paper includes an empirical evaluation for surface reconstruction on DTU and novel view synthesis on the Mip-NeRF360 dataset.
Furthermore, the authors ablate on the choice and weightings of depth smoothness and connectivity / scene loss terms.

**Strengths:**

- The paper is well written and very easy to follow and understand.
  - The introduction motivates the task and the proposed approach well.
  - The related work is very detailed and covers many existing approaches.
  - The method is is mostly clearly described.
  - The paper is honest about its experimental results.
- The paper includes multiple technical contributions:
  - The RTS representation based on triangle primitives intuitive.
    - Colors are interpolated based on the Spherical Harmonic coefficients of the triangle's vertices.
    - Surface normals per triangle are computed based on cross-products between edges and then alpha blended.
  - I find the idea of the connectivity (scene) loss to encourage emergence of closed surfaces consisting of aligned triangles as in meshes especially interesting and novel.
  - The adaptive density control leverages the characteristics of the representation, e.g., by splitting triangles into 4, or making use of the edge connectivity for pruning criteria.
- The method achieves the best performance in 3D surface reconstruction for some scenes of DTU and the best LPIPS result in novel view synthesis on indoor scenes of the Mip-NeRF360 dataset.
- The ablation study w.r.t. loss terms and weighting shows the effectiveness of the connectivity loss.
- The appendix provides additional implementation details as well as qualitative and quantitative results.

**Weaknesses:**

- The quantitative comparison with baselines is not convincing.
  - PGSR [1] is overall better in both surface reconstruction (Tab. 1) and novel view synthesis (Tab. 2) than the proposed method.
    - The authors attribute this to "the algorithm utilizes a full suite of multi-view objective functions that significantly improve the geometric reconstruction quality" (line 443). It remains unclear (also from related work) what these technical differences are exactly and whether comparison is still fair or not.
    - In any way, what prevents the authors from using the same objective functions for a fair comparison with PGSR?
  - While on indoor scenes, the method's NVS performance is on par with baselines, the performance on outdoor scenes is quite poor, lacking significantly behind GOF [2] and PGSR [1] (3.41 less PSNR and 0.147 higher LPIPS).
- The experimental evaluation is limited.
  - The provided qualitative results are insufficient.
    - The main paper includes only Fig. 7, which does shows three DTU examples without any comparison to baseline methods, and Fig. 8, which shows one novel view each for two DTU examples compared to only one baseline 2DGS, which is quantitatively only the 3rd strongest baseline for surface reconstruction, according to Tab.1.
  - The paper misses to compare their method with baselines in terms of training and rendering speed as well as GPU memory requirements.
    - The authors mention in the limitations (Sec. 6) that "due to periodic nearest-neighbors search, there is a minor increase in run-time".
- The paper lacks a detailed comparison of their method with TriangleSplatting [3].
  - This paper seems to be an extremely important related work, but the paper does not mention it at all in the introduction and mentions it in one sentence in the related work (line 146f.).
  - It is very difficult to evaluate the technical novelty compared to this paper, especially regarding the proposed representation, as the authors inly mention that TriangleSplatting "does not support any mechanism for [the triangles] to interact directly with each other", i.e., claiming that their soft connectivity forces loss is novel.
- Lack of clarity:
  - The authors claim that previous approaches like 3D Convex Splatting [4] are "more expensive than RTS" (cf. lines 153-163). However, the difference to RTS here is not clear to me, especially since in terms of diffuseness of primitives this method seems to share a lot of similarities with 3DCS (cf. 181 and Sec. 3.3). Furthermore, as mentioned above already, the paper misses to evaluate efficiency in terms of time and memory and compare with baselines to support this claim.
  - The triangles are parameterized by the three bisector lengths. It would be helpful for the reader to mention that a triangle is indeed uniquely represented by this.
  - The initialization of the rotation matrix is unclear. First, the rotation is defined based on triangle edges and normals, but then you apply a random rotation to this matrix. Is the outcome then not just random too?
  - "The diffuse scalar is also set as a function of the average distance to the three nearest neighboring points." (line 212). What is the function? Or point to the appendix, if it is described there.
  - In Sec. 3.5, the authors first describe the intuition and the behavior of the soft connectivity forces they introduce without actually defining the loss term. This is done later in Sec. 4.2. These two section should be merged to improve readability.
   - The description of the second term in the connectivity loss (the normal part) is missing in line 356.

- Minor weaknesses:
  - Missing references for use of triangles as fundamental primitives in computer graphs (lines 58f.).
  - The Fig. 4 is unnecessarily large / has a suboptimal layout for its message.
  - Missing references: Which previous works do you refer to in line 209 "Similar to previous works...".

References:
- [1] PGSR: Planar-based Gaussian Splatting for Efficient and High-Fidelity Surface Reconstruction. TVCG 2024
- [2] Gaussian Opacity Fields: Efficient Adaptive Surface Reconstruction in Unbounded Scenes. SIGGRAPH Asia 2024
- [3] Triangle Splatting for Real-Time Radiance Field Rendering. arxiv 25 May 2025
- [4] 3D Convex Splatting: Radiance Field Rendering with 3D Smooth Convexes. CVPR 2025

**Questions:**

- The authors need to compare their method with TriangleSplatting in terms of technically novel contributions.
- I also suggest that the authors either describe in detail what they mean with PGSR's "algorithm utilizes a full suite of multi-view objective functions" as the reason for their worse performance compared to PGSR, why it is not applicable to their approach, if that is the case, or provide additional experimental results for using the same objective functions as PGSR but with the RTS representation and connectivity loss, hopefully boosting performance.
- Since the authors claim that previous works relying on volumetric primitives are "more expensive than RTS" (line 163), I suggest that the authors provide empirical evidence for this in form of an time and memory efficiency comparison for both training and test time.
- Further open questions are:
  - What is the intuition of applying a random rotation to the rotation matrices of triangles at initialization? Is it maybe a rotation only a long a certain axis?
  - The authors emphasize that they "directly generate a 3D point cloud for geometric evaluation from the rendered depth maps without performing any TSDF fusion" (line 405f.). Is this consistent with baselines? If not, how does it change the conclusion of the comparisons?

---

> ### Author Response · Authors · 2025-11-22
>
> First and foremost, we would like to thank the reviewer for their time and for their feedback and comments on our work. We appreciate the detail provided in the response.
>
>
> ## Weaknesses
> ### (1a) " It remains unclear (also from related work) what these technical differences are exactly [w.r.t PGSR [1]] and whether comparison is still fair or not..."
> PGSR is indeed related to our method, with many technical differences mainly due to the choice of primitive as well as the additional loss functions. Their framework applies heavy supervision across multiple views by rendering nearby images and enforcing reprojection criteria during optimization. These losses are indeed applicable to our framework; however, it was our intention to present this entirely new approach using triangle primitives, and a primitive-to-primitive loss, without implementing all of the algorithmic options afforded to methods dealing in Gaussian kernels, with the assumption that these algorithmic additions can be applied to RTS in future work.
>
> Along these lines, the intended baseline for the most direct comparison with our method is 2DGS. Employing the same SSIM and Normal Consistency losses, our use and parameterization of triangle primitives directly leads to improvements in geometric reconstruction of surfaces. This claim is supported by the first row in Table 3, in which we evaluate our approach using the same losses as 2DGS (without our additional losses), measuring 0.67 overall Chamfer distance on the DTU dataset compared to 0.80 for 2DGS (from Table 1).
>
> Furthermore, to remain consistent with previous literature, the evaluation in Table 1 utilizes a ground truth foreground-background segmentation mask. This mask is not provided by the DTU dataset and is not a part of the official benchmark evaluation, since its use significantly simplifies the reconstruction. In order to portray a more fair evaluation of reconstruction quality, we provide a comparison evaluating 2DGS, PGSR, and RTS with and without this ground truth mask (following the official DTU benchmark) in Table S.2 of the supplemental material. The Chamfer distance degradation is as follows: (i) 2DGS is 0.80 to 1.15; (ii) PGSR is 0.52 to 1.04; (iii) RTS is 0.61 to 0.82. We show that even though the reconstruction quality for all methods decreases, RTS is much more robust to the removal of the ground truth masks, as our method handles floating primitives much more effectively than the baseline methods.
>
> This motivates the need for direct losses on the primitives in 3D, whereas the multi-view losses applied in PGSR have little effect in regions of the scene with inconsistent geometry, which ultimately get filtered out by the ground truth masks.
>
>
> ### (1b) "what prevents the authors from using the same objective functions for a fair comparison with PGSR?"
> Please see the response to **Weakness (1a)** above.
>
>
> ### (1c) "While on indoor scenes, the method's NVS performance is on par with baselines, the performance on outdoor scenes is quite poor..."
> Much of the outdoor scenes in the mip-NeRF 360 dataset include dense background foliage. This is a notable limitation of the use of primitives with sharp boundaries. However, the foreground regions of the scene are very precisely captured with our representation, even better than by many Gaussian kernel-based methods. This is not reflected in the PSNR metric, which favors blurred renderings and penalizes high-frequency ones. We will provide further qualitative comparisons of this effect in the supplemental material in a revised version of the paper.

---

> > ### Author Response · Authors · 2025-11-22
> >
> > ### (2a) "provided qualitative results are insufficient..."
> > We will provide more qualitative comparisons to the baselines in a revised version of the paper.
> >
> > ### (2b) "The paper misses to compare their method with baselines in terms of training and rendering speed as well as GPU memory requirements..."
> > We provide all implementation details, as well as run-time and primitive count information in the supplemental material in lines 779-794. The run-time for our implementation is 1.5h and 4.5h with 249,941 and 1,275,985 primitives on the DTU and mip-NeRF 360 datasets, respectively.
> >
> > We initially chose to only report on our run-times and not directly report the run-times of other methods due to some major discrepancies in the reported run-times of related works across different publications. For example, the PGSR publication reports run-times of 0.32h and 2h for 2DGS and GOF on the DTU dataset, respectively. While the GOF publication reports run-times of 0.18h and 0.31h for 2DGS and GOF on the DTU dataset, respectively. Run-time is highly dependent on the implementation of the algorithm and the hardware it is executed on.
> >
> > Additionally, the code of our implementation is an entirely new approach with many custom CUDA kernels and is not yet fully optimized. For completeness, we report all of this information for our implementation on our hardware, and for clarity, we specify the impact the additional scene loss has on these numbers, which we provide explicitly in lines 779-789 in the *Edge Connectivity Overhead* paragraph, in which we explain that the minimal overhead leads to roughly a 6% increase in run-time.
> >
> > We can append these values (taken directly from each publication) to the tables in the experiments; however we do not feel this would be a meaningful contribution.
> >
> > ### (3) "The paper lacks a detailed comparison of their method with TriangleSplatting [3]..."
> > TriangleSplatting is an unpublished work that is concurrent with our own. As the reviewer mentions, we reference this paper as a part of related work and provide a quantitative comparison against this work in our evaluations, in which the authors kindly provided us with their geometric results.
> >
> > RTS shows a major increase in geometric accuracy compared to TriangleSplatting. As we mention in the section on related work, TriangleSplatting use translucent triangles as their primitive. Each triangle in their work is parameterized by a set of 3D points, similar to their previous work in 3D Convex Splatting. The primitives in our work are parameterized by an incenter, scalar offsets along the interior angle bisectors, and a rotation quaternion. Parameterizing the primitives in these separate terms allows for a more selective propagation of gradients with different learning rates, which in our experimentation, showed more favorable convergence toward a higher quality reconstruction. The activation function used in TriangleSplatting also differs from ours, as they use the normalized ReLU of the SDF for each triangle, whereas we apply a Sigmoid function to the SDF for each triangle. Both works have very similar motivations, in which using triangles as the primitive in an alpha-blending optimization framework provides a direct route to the estimation of a mesh with high-quality appearance that can directly be rendered in novel views.
> >
> > We agree with the reviewer, and will modify the text adding a more detailed description of the related method.

---

> > > ### Author Response · Authors · 2025-11-22
> > >
> > > ### (4a) "The authors claim that previous approaches like 3D Convex Splatting [4] are "more expensive than RTS"..."
> > > Since the underlying algorithm of 3DCS relies on the projection of six points per-primitive into the image plane, followed by the computation of several 2D signed distances and several activation functions when performing rasterization, the algorithm itself is less efficient than rasterizing a single triangle with three vertices and 3 activation functions.
> > >
> > > With that being said, we agree that this comment is not properly supported by the experiments and thus will be removed from the paper.
> > >
> > > ### (4b) "The triangles are parameterized by the three bisector lengths..."
> > > We will add this comment for clarity.
> > >
> > > ### (4c) "The initialization of the rotation matrix is unclear..."
> > > The initial orientation of the triangle must be established w.r.t each individual vertex when it is created. The comments in the paper describe the choice of canonical orientation before applying a random rotation to each primitive.
> > >
> > > This is a description of the actual implementation, which is important to know when splitting the triangles. We can move this description into the implementation details in the supplemental material, and specify only that the initial rotation of each primitive is random in the main paper.
> > >
> > > ### (4d) " 'The diffuse scalar is also set as a function of the average distance to the three nearest neighboring points.' (line 212). What is the function? Or point to the appendix, if it is described there..."
> > > The diffuse scalar is similar to the scale values, in which it is directly proportional to the average distance to the three nearest neighbors. We can provide a reference to the actual function in the supplemental material. This is not critical to the paper, since changing this function in a reasonable way did not significantly affect the final results.
> > >
> > > ### (4e) "In Sec. 3.5, the authors first describe the intuition and the behavior of the soft connectivity forces they introduce without actually defining the loss term. This is done later in Sec. 4.2. These two sections should be merged to improve readability..."
> > > We appreciate the feedback on readability from the reviewer and will ask feedback from our peers on the clarity and flow of these two sections.
> > >
> > > ### (4f) "The description of the second term in the connectivity loss (the normal part) is missing in line 356..."
> > > We will add the intuition behind the second part of this loss term.
> > >
> > >
> > > ## Minor Weaknesses
> > > ### (1) "Missing references for use of triangles as fundamental primitives in computer graphs (lines 58f.)..."
> > > We will add this reference in the introduction.
> > >
> > > ### (2) "The Fig. 4 is unnecessarily large / has a suboptimal layout for its message..."
> > > We were concerned that this figure may not appear clearly if it was made too small. We will take this feedback into account.
> > >
> > > ### (3) "Missing references: Which previous works do you refer to in line 209 'Similar to previous works'..."
> > > This refers to most Gaussian Splatting implementations. We will use 3DGS as the main reference here.
> > >
> > > ## Questions
> > > Since the questions are a summary of the above weaknesses, please refer directly to the above responses.

---

### Author Response · Authors · 2025-12-03
**Modification Report to AC/Reviewers**

In response to comments by all reviewers, we have made the following modifications to our paper. Please use the lists below to locate each change in our recent revision. To assist in reviewing, we use the color magenta to identify all modified/new text and all modified/new figures.

We have made modifications in the following parts of the main paper:
1. Lines 162-164 in Section 2
2. Lines 205-206 in Section 3.1
3. Lines 215-216 in Section 3.2
4. Lines 361-363 in Section 4.2

We have made modifications in the following parts of the Supplemental Material:
1. Added Table S.1 in Lines 757-764
2. Added Equation S.4 in Lines 768-775
3. Added Lines 791-792
4. Added Related Work extension in Lines 793-843
5. Updated images for Figure S.1
6. Added Figure S.2
7. Added Lines 904-906
8. Added Figure S.3

---

### Meta-Review · Area_Chair_WbR6 · 2025-12-29

**Summary:**

The authors present a framework for triangle soup optimization with alpha blending and added soft connectivity constraints that encourage triangle edges to become connected. The work was originally received with negatively leaning reviews (2, 2, 6, 8), where the main concerns evolve around unconvincing results and limited novelty (see below). During the discussion, the authors provided explanations and an additional qualitative results but where not able to convincingly rebut the reviewers concerns. I follow with a reject recommendation at this point.

For a resubmission, I encourage the authors to focus on their core contribution in this work, enforcing soft connectivity between triangles in the optimization procedure, instead of introducing a whole new framework (which is very similar to previous works). Then, more effort needs to be made to exhaustively evaluate this contribution on top of the state of the art qualitatively and quantitatively, clearly showcasing the benefits of the addition.

**Reviewer Concerns:**

*1) Quantitative results not competitive with the state of the art.* This was not sufficiently addressed by the authors. They point out that in comparison to the 2DGS baseline, which uses a similar set of losses, a gain in reconstruction quality was achieved. However, I believe more effort needs to be made to combine the method with recent approaches to see if a joint method can achieve state of the art quality, i.e. if the presented contribution of triangle attraction can add additional benefits on top of them. Also, it is concerning that the NVS quality barely reaches that of NeRF.

*2) Limited experimental evaluation.* Only a small amount of qualitative results - the authors added more results in the discussion. Also, missing comparisons on runtime and optimization speeds against previous approaches. In a saturated field like this, I believe it would be important to fairly compare the methods on the same hardware - not addressed.

*3) No comparison against TriangleSplatting.* The authors correctly pointed out that this is technically a concurrent work. However, paper and code where released 2-3 month before the ICLR deadline. This will not be considered anyway.

*4) Clarity concerns (normal loss and others).* The authors addressed them / promised to address in the discussion.

*5) Limited Novelty.* The reviewers point out that triangle optimization is not novel but has been done by several previous works. The main novelty is the soft connectivity between triangles. The authors agree and did not state otherwise in the paper.

**Reviewer Scores:**

Reviewers wzMp and TvUE gave score 2 in the initial reviews, recommending reject. Their main concerns are evolving around unconvincing quantitative and qualitative results and limited novelty compared to previous works on triangle set optimization. Since the authors address these concerns only sparsely, I don't believe their scores would have significantly moved into the positive direction.

Reviewers avbo and QKuE already started off with positive recommendations (8,6) and would have probably remained with their scores as well.

---

### Decision · Program_Chairs · 2026-01-26

Reject